

# Uncertainties in the national inventory of methane emissions from rice cultivation: field measurements and modeling approaches

Wen Zhang[1], Tingting Li[1], Wenjuan Sun[2*]

1 LAPC, Institute of Atmospheric Physics, Chinese Academy of Sciences, Beijing, China

2 LVEC, Institute of Botany, Chinese Academy of Sciences, Beijing, China

*Correspondence author, Email: sunwj@ibcas.ac.cn

## Abstract

Uncertainties in national inventories originate from a variety of sources, including methodological failures, errors and insufficiency of supporting data. In this study, we analyzed these sources and their contribution to uncertainty in the national inventory of rice paddy methane emissions in China and compared the differences in the approaches used (e.g., direct measurements, simple regressions and more complicated models). For the 495 field measurements we collected from the scientific literature, the area-weighted 95% CI ranged from 13.7 to 1115.4 kg $CH_4$ $ha^{-1}$, and the histogram distribution of the measurements agreed well with parameterized gamma distributions. For the models, we compared the performance of methods of different complexity (i.e., the CH4MOD model, representing a complicated method, and two less complex statistical regression models) to evaluate the uncertainties associated with model performance as well as the quality and accessibility of the regional datasets. Comparisons revealed that the CH4MOD model performed better than the comparatively simple regression models only when sufficient input data for the model were available, with the regression equations performing better otherwise. As simulated by CH4MOD, methane fluxes varied from 17.2 kg $CH_4$ $ha^{-1}$ to 708.3 kg $CH_4$ $ha^{-1}$, covering 63% of the range of the field measurements. When applying the modeling approach to the 10 km × 10 km gridded dataset of the model input variables, within-grid variations were found to represent 81.2% – 95.5% of the modeled mean fluxes. Moreover, up-scaling the grid estimates to the national inventory resulted in the models contributing 56.6% of the total uncertainty, with the remaining 43.4% being attributed to errors and the scarcity of the spatial datasets of the model inputs. Our analysis reveals the dilemma between model performance and data availability



5    when using a modeling approach: a model with better performance may help in reducing uncertainty caused by model fallacy but increases the uncertainty caused by data scarcity, as greater levels of input are needed to improve performance. Reducing the total uncertainty in the national methane inventory depends both on a better understanding of the complexity of the mechanisms of methane emission and the

10    spatial correlations of the factors that influence methane emissions from rice paddies.

**Keywords**: Uncertainty, source and contribution, spatial variation, national inventory, methane emission



## 1 Introduction

Rice cultivation is a major source of anthropogenic methane and a prime target of greenhouse gas mitigation efforts (Smith et al., 2008). Numerous methods have been applied for estimating national and global inventories of rice paddy methane emissions, including meta-analysis of direct measurements, process models and empirically based statistical models. However, the range of regional/global source estimates remains large (Cao et al., 1996; Sass et al., 1999; Chen et al., 2013). The major factors that are known to regulate rice paddy methane emissions include agricultural management practices (Khosa et al., 2011; Sanchis et al., 2012; Sass et al., 1992; Bodelier and Laanbroek, 2006) and environmental conditions, such as climate and soil properties (Conrad et al., 2007; Inubushi et al., 2011; Sass et al., 1991). Currently, techniques for calculating methane emissions differ substantially and are far from perfect, mainly because of issues involving methodological fallacy and data insufficiency.

By extrapolating field measurements obtained from experiments, methane emissions from the 30 million hectares or so of land under rice cultivation in China were estimated to range from 21.6 Tg $CH_4$ $yr^{-1}$ to 30 Tg $CH_4$ $yr^{-1}$ (Matthews et al., 1991; Taylor et al., 1991). The extrapolation of methane emission rates from site measurements to larger regions is unlikely to yield reliable results because of the tremendous spatial heterogeneity in environmental conditions and agronomic activities (Ogle et al., 2010). Other studies have described the relationships between methane emissions and rice NPP (net primary productivity) (Bachelet and Neue, 1993) and organic matter inputs (Bachelet et al., 1995). Ambient temperature and the use of nitrogen (N) fertilizer have also been identified as determinants of methane emissions (Kern et al., 1995; Bachelet et al., 1995). Until the significant reduction in methane emissions caused by mid-season drainage was confirmed (Sass and Fisher, 1997;Yagi et al., 1997;Li et al., 2002;Yan et al., 2005), all previous regional and national estimates (obtained using extrapolation or regression equations) were derived from continuously flooded rice fields. More recently, factors such as the rice cultivar involved (Watanabe et al., 1995; Butterbach-Bahl et al., 1997; Ding et al., 1999; Inubushi et al., 2011), soil properties (Sass et al., 1994; Yao et al., 1999) and atmospheric ozone concentrations (Bhatia et al., 2011) have been incorporated into models designed to estimate methane emissions from rice paddies. Complex



interactions among these factors have spurred model development (Cao et al., 1995; Li, 2000; Matthews et al., 2001; Huang et al., 1998; Van Bodegom et al., 2001; Huang et al., 2004). To delineate variations in methane emissions and to reduce uncertainties, the impacts of these factors on the production, oxidation and emission of methane were mathematically incorporated into the models. Models with a greater

number of factors involved are able to reduce uncertainties in estimating methane emissions, but the estimates generated by these models still differ significantly across multiple spatial and temporal scales (Butenhoff et al., 2009; Ren et al., 2011; Chen et al., 2013).

Reduction of the uncertainty in estimated methane emissions requires the

development of an effective and reliable model that incorporates various paddy environments and agronomic activities. However, our understanding of the complex biogeochemical processes that occur in paddy soils is poor. When estimating methane emissions from rice agriculture, only factors that are thought to be key determinants of methane emissions have been incorporated into the models. Excluding "non-key"

factors introduces errors into the model output (Equations C2 and C3 in the Supporting Information). Improving our knowledge of methane processes in the future will increase the number of factors that are integrated into models and potentially delineate details related to spatial/temporal variations.

Uncertainties in regional estimates of methane emissions from rice paddies stem not

only from deficiencies in the applied models but also from errors and inadequate data, which we discussed in a previous study (Zhang et al., 2014; Appendix C in the Supporting Information). A model with more factors generally performs better than a model with fewer factors but requires a larger amount of data to facilitate model performance. A model with good performance can still result in large uncertainties

when the available input data (e.g., soil properties, rice irrigation, types and amount of organic matter) are insufficient (Zhang et al., 2014).

In the present study, we analyzed the uncertainties in experimental measurements of methane fluxes in different rice paddies. We also evaluated the performance of different methods involving a diversity of input variables and the

influence of data availability on the performance of these methods. Finally, the uncertainty in the national emissions inventory as a consequence of variable model performance and according to the quality and availability of input data were discussed.



## 2 Materials and methods

### 2.1 Field measurements of methane emissions from rice paddies in China

The observational data used in this study (Table 1) consisted of field methane fluxes measured at 33 sites (Fig. 1). We obtained these measurements from the published literature concerning all crop rotations with rice cultivation in China (double rice, winter wheat and rice rotation, single rice crop cultivation, and so forth) (Wei, 2012). A total of 495 measurements were taken at the 33 sites. The amount of organic matter added to the rice paddies ranged from 0 t C ha$^{-1}$ to 15.3 t C ha$^{-1}$ and included animal manure, green manure, crop straw, biogas residuals and their various components. The applied water regimes consisted of continuous flooding, single mid-season drainage and multi-drainage irrigations.

Model performance was assessed by comparing the model estimates with the measurements. To drive the models, data pertaining to rice yields, soil properties and crop phenologies were collected from the relevant literature (Appendix B in the Supporting Information).

### 2.2 Performance of the methods used to estimate methane emissions

The uncertainties produced by the models derive from inaccuracies in the models themselves (Appendix C in the Supporting Information) as well as from the quality and availability of data (Fig. 2). Model performance was assessed by comparing model outputs with the direct measurements (left part in Fig. 2). Errors in the input data of the model can be propagated in the obtained estimates (right side of Fig. 2, Appendix D in the Supporting Information).

Many techniques are available for calculating estimates of rice paddy methane emissions, such as extrapolation of measured emission rates (Khalil et al., 1991; Khalil et al., 1993), statistical regression equations (Bachelet et al., 1995; Kern et al., 1995; Kern et al., 1997) and the application of models of varying complexity (Cao et al., 1995; Matthews et al., 2001; Van Bodegom et al., 2001; Huang et al., 1998; Li, 2000). We compared the performance of these methods under different levels of data availability (Table 1) using experimental field measurements as a point of reference (Fig. 1). In Table 1, R1 represents a simple regression equation in which the carbon (C) input is the sole predictor (Neue et al., 1990). Regression equation R2 is slightly more complicated in that it uses organic C and fertilizer N application as inputs (Kern




et al., 1997). We assumed two data availability scenarios for R2. In R2－S0, both the C and N inputs are available; in R2－S1, only the C input is available (Table 1).

The third approach consists of a semi-empirical model, CH4MOD. This model was developed to simulate methane emissions from rice paddies under diverse environmental conditions and various agricultural practices (Huang et al., 1998; Huang et al., 2004). The input variables of the model include the climate, soil conditions, water management type, organic matter application and crop rotations. The model consists of two modules: the derivation of methanogenic substrates from added organic matter and rice root exudates and the production and emission of methane. Rice biomass is a key variable used to calculate the root exudates and the fraction of the methane emitted by rice plants and bubbles. The daily changes in the soil redox potential (Eh) were calculated according to various water manipulations conducted in the rice paddies (Xie et al., 2010b). The influences of other environmental factors, such as soil temperature and texture, on the decomposition of organic matter and the production of methane were expressed as specific coefficient functions (Huang et al., 1998). The input variables of the CH4MOD model (Appendix B in the Supporting Information) include the daily air temperature, soil sand percentage ($SAND$), organic matter amendment ($OM$), rice grain yield ($GY$), water management pattern ($W_{ptn}$) and rice cultivar index ($VI$).

Four model input scenarios (Table 1) were scheduled to evaluate the performance of CH4MOD under different levels of data availability. In M－S0, all of the model variables were assigned specific values. In M－S1, the application of organic matter was assigned the average value for all experiments, thus assuming a situation where no detailed information on organic matter application was available. In M－S2, detailed information on the water regime and soil properties was assumed to be unavailable. In M－S3, detailed information on all three major factors (organic matter application, soil properties and water regime) was assumed to be unavailable.

The estimation residuals (Δy, Equation 1), bias ($B_R$, Equation 2) and coefficient of variations ($CV$, Equation 3) were thus evaluated as follows:

$$\Delta y_i = \hat{y}_i - y_i, \qquad i = 1,2,\dots,N \tag{1}$$

$$B_R = \frac{E(\Delta y)}{m} \times 100\% \tag{2}$$

$$CV = \frac{\sqrt{E((\Delta y)^2) - (E(\Delta y))^2}}{m} \times 100\% \tag{3}$$



where $y$ represents the measured methane fluxes; $\hat{y}$ is the estimate of $y$; and $N$ is the total number of measurements. $E(\ )$ indicates the statistical mean, and $m = E(y)$ is the mean of the measured methane fluxes ($y_i$). The total error of the estimation is calculated as follows:

$$ERR_T = m \times \sqrt{B_R{}^2 + CV^2} \tag{4}$$

## 2.3 Uncertainties in estimating rice paddy methane emissions on regional scales: data error and availability

In addition to errors in the performance of the methods, the difficulties in estimating regional rice paddy methane emissions also stem from errors in, and limited availability of, input data. To measure the uncertainties in model outputs due to insufficient data quality and availability, we applied Monte Carlo simulations (Penman, 2000) to the CH4MOD model. Statistical characteristics were derived from the available datasets to develop probability distribution functions (PDFs) for each model input variable. At the last step of the Monte Carlo simulation, the mean and the 95% CI of the methane flux were derived from the iterations of the CH4 MOD simulation (Zhang et al., 2014).

The factors involved in the uncertainty analysis included organic matter application, soil properties and water regimes; these variables ($OM$, $SAND$, and $W_{ptn}$) were parameterized as input variables in the CH4MOD model (Huang et al., 2006; Zhang et al., 2011). The other two model input variables were the rice grain yield and daily ambient air temperature. These two variables were not used in the uncertainty analysis because sufficient relevant data were available, which were characterized by less error compared with the other variables (Zhang et al., 2014).

The $SAND$ data were obtained from a 10 km $\times$ 10 km grid dataset interpolated from soil survey data (Oberthür et al., 1999;Shi et al., 2004;Liu et al., 2006). It is possible that approximately half (Van Bodegom et al., 2002b) of the immense spatial variation in soil properties can be lost after spatial interpolation (Goovaerts, 2001); as a result, the missing spatial variation was interpolated from the PDF of the gridded $SAND$ data.

The organic matter inputs in the rice fields consisted of various types of farm manure (green manure and animal feces), crop straw, and dead roots and stubble leftover from previous harvests. Root and straw biomass were calculated using the root/shoot ratio and harvest indices (Huang et al., 2007; Gao et al., 2002; Xie et al.,



2010). Stubble was assumed to represent one-tenth of the straw biomass. The proportions of incorporated straw and applied farm manure were derived from data obtained from two large-scale investigations, the First National Census of Pollution Sources conducted by China's Ministry of Environmental Protection (CFPC, 2011) and censusing conducted by the Institute of Atmospheric Physics, Chinese Academy of Sciences. The proportion of straw and the amount of manure incorporated into the crop fields were summarized by province (Table B1 in the Supporting Information).

Data pertaining to $W_{ptn}$ were only very rarely available on a regional scale. The limited information provided in a few studies (Mao, 1981; Liang, 1983; Xiong et al., 1992; Cai et al., 2003; Ma et al., 2005; Ministry of Water Resources and Utilization of China (MWRUC), 1996) enabled us to produce rough estimates related to irrigation in regions of major rice cultivation (Table B2 in the Supporting Information).

The data pertaining to the rice grain yield and harvesting area as of 2005 were obtained from China's Statistical Yearbook (EBCAY, 2006) and the nation's agricultural database maintained by the Chinese Academy of Agricultural Sciences, respectively. The spatial distributions of all rice paddies in 2005 and the rice paddy area within each 1 km × 1 km grid were obtained from the Data Center for Resources and Environmental Sciences of the Chinese Academy of Sciences (RESDC, CAS). Daily mean air temperature data from 678 meteorological stations throughout China for 2005 were acquired from the National Meteorological Information Center (NMIC) of the China Meteorological Administration (CMA) (http://cdc.cma.gov.cn/). The temperatures were then spatially interpolated into 10 km × 10 km grids for each day according to the method described by Thornton et al. (1997). Details on the datasets used in this study can be found in Appendix C.

To preserve details related to spatial variations, all data input into the model were converted into 10 km × 10 km grids. The applied rasterization techniques and details of how the model was run on raster datasets were provided in previously published papers (Huang et al., 2006).

## 2.4 Combining uncertainty and spatial aggregation

In each 10 km × 10 km grid, the uncertainties in our estimates originated from errors in both model performance and the input data. Equation 5 was used to merge the two components (Appendix D in the Supporting Information):





$\sigma_{T,j}{}^2 = ((F_j \times B_R)^2 + (F_j \times CV)^2) + \sigma_{D,j}{}^2$            (5)

where $\sigma_{T,j}$ represents the uncertainty of the methane flux in grid $j$, and $F_j$ and $\sigma_{D,j}$ represent the mean and standard deviation of the Monte Carlo simulation results in grid $j$, respectively. $B_R$ and $CV$ represent the same entities as in Equations 2 and 3.

To produce the uncertainty of the national inventory, the three components of the

estimation uncertainties (Equation 5) in all grids were separately aggregated (Equation D2) and summed (Equation 6) as follows:

$$\sigma_T{}^2 = \sigma_{BR}^2 + \sigma_{CV}^2 + \sigma_D^2$$            (6)

## 3 Results

### 3.1 Methane emissions and the uncertainties derived from field measurements

Among the 495 methane flux measurements, 184 (37% of all cases) came from paddies that were continuously flooded during the entire rice growing period; 50 (10% of all cases) came from paddies with single mid-season drainage; and 261 (53% of all cases) came from paddies under multi-drainage. The average methane fluxes associated with the three water regimes were $531.6 \pm 512.6$, $251.6 \pm 231.1$ and $224.1$

$\pm 207.5$ kg CH$_4$ ha$^{-1}$, respectively (Fig. 3a). The overall arithmetic average of the 495 measurements (represented hereafter by $m_c$) was $341.2 \pm 383.2$ kg CH$_4$ ha$^{-1}$. However, the simple arithmetic average might be a biased representation of the "true" mean methane flux of rice paddies in China, as far less than 37% of the rice paddies in China are continuously flooded. In the literature, 10%, 20% and 70% of the rice area

was reported to be under continuous flooding, single drainage and multi-drainage water regimes, respectively (Xiong et al., 1992;Ministry of Water Resources and Utilization of China (MWRUC), 1996), and the area-weighted mean (Appendix A in the Supporting Information) of the measured fluxes (represented hereafter by $m_w$) was $260.4 \pm 281.6$ kg CH$_4$ ha$^{-1}$ (Fig. 3a).

The 95% confidence intervals (CIs) of the methane flux measurements were $61.1$–$2,145.9$ kg CH$_4$ ha$^{-1}$, $9.6$–$809.9$ kg CH$_4$ ha$^{-1}$ and $14.0$–$797.7$ kg CH$_4$ ha$^{-1}$, respectively, for the three water regimes (Fig. 3a). The 95% CI of all combined area-weighted measurements (Appendix A in the Supporting Information) was $13.7$–$1,115.4$ kg CH$_4$ ha$^{-1}$. The measurements were not normally or symmetrically

distributed (Fig. 3b). The P-P plots (Fig. 4) showed that the parameterized gamma distributions matched the sample distributions. The 95% CIs calculated with the





parameterized gamma functions were 16.8–1,900.8 kg $CH_4$ ha$^{-1}$, 10.4–863.4 kg $CH_4$ ha$^{-1}$ and 8.9–774.2 kg $CH_4$ ha$^{-1}$, respectively, for the three water regimes; these values overlapped the CIs derived directly from the measurements by 88.2%, 99.9% and 97.0%, respectively.

The national methane emissions from rice agriculture calculated by multiplying

the rice harvesting area (yearbook data in 2005) by the area-weighted mean flux (260.4 ± 281.6 kg $CH_4$ ha$^{-1}$) were 7.51 Tg $CH_4$ (Fig. 3a). The uncertainty of the national emissions is usually represented by the standard error (SE). When the measurements are statistically independent, SE is $n-1$ ($n$ is the sample size of the measurements) times smaller than the standard deviation (± 281.6 kg $CH_4$ ha$^{-1}$),

which consists of the representative and measurement errors of the measured fluxes (Van Bodegom et al., 2002a; Verburg et al., 2006). Assuming that the measurements were statistically independent, the 95% CI of the national inventory was 7.20–8.58 Tg $CH_4$ (Equation A3 in the Supporting Information). However, the independency assumption is questionable because of the correlations between the spatially

correlated background environmental conditions and agricultural activities (Legendre, 1993; Dormann et al., 2007). The equivalent sample size, $n$, used to calculate SE may be smaller than 495, and the 95% CI of the national inventory is therefore larger than that with the independency assumption.

### 3.2 Model performance under different situations of data availability

R1 was more likely to overestimate the amount of methane emitted than R2 (Table 2), especially when more organic matter was incorporated (Fig. 5a). The estimated methane flux calculated by R1 was more than 6,000 kg $CH_4$ ha$^{-1}$, whereas the corresponding measured methane flux was less than 3,000 kg $CH_4$ ha$^{-1}$ (Fig. 5a). The averaged bias of the estimate obtained with R1 was 212.0 kg $CH_4$ ha$^{-1}$ (Table 2)

or 62.1% of the measured mean ($m_c$ = 341.2 kg $CH_4$ ha$^{-1}$). The average bias of R2, in contrast, was −1.3 kg $CH_4$ ha$^{-1}$; the estimates obtained using R2 did not show significant variations and appeared to decline when the measured methane fluxes increased (Fig. 5b). The CH4MOD model also produced a small averaged bias, representing 7.1% of the measured mean. The total estimation errors were 253.0,

407.8 and 596.0 kg $CH_4$ ha$^{-1}$ for the M－S0, R2－S0 and R1－S0 scenarios, respectively (Table 2), which demonstrates that model performance improves when more factors are incorporated into the model.





Although the CH4MOD model produced better simulation results than the simple regression equations, its performance fundamentally depends on data availability. When no case-specific data were available (as in scenario M－S3), $B_R$ was −32.2%, and $ERR_T$ was 122.1% of the mean flux; the results obtained under this scenario were even worse than the results obtained under the R2－S0 scenario (Table 2). For the M－S1 scenario, where the data pertaining to the soil properties and water regime were case-specific, the magnitude of $B_R$ decreased to 9.0% of the mean flux, and the total error decreased to 101.2% of the mean flux. The M－S0 scenario produced much better results than the other scenarios, as more data were available for the key model input variables (Table 2).

**3.3 Inventory of rice paddy methane emissions and spatial variations**

Because of the spatial heterogeneity in the climate, soil properties, organic matter incorporation and field irrigation in rice cultivation, the simulated methane fluxes varied spatially between 17.2 kg $CH_4$ $ha^{-1}$ and 708.3 kg $CH_4$ $ha^{-1}$ from grid to grid (Fig. 6). The national means for the simulated methane fluxes were 217.9 kg $CH_4$ $ha^{-1}$, 204.6 kg $CH_4$ $ha^{-1}$ and 255.8 kg $CH_4$ $ha^{-1}$ for single, early and late rice cultivation, respectively. The within-grid estimation error ($\sigma_{T,j}$, calculated with Equation 5) represented 81.2% – 95.5% of the mean fluxes, $F_j$, in the grids.

As shown in Fig. 7, the highest levels of emitted methane occurred in southern China, with the northeast also representing a major source of methane, despite this region being climatically cool. The total amount of methane emitted, as calculated by the modeling approach, was 6.43 (3.79 – 9.77) Tg (Table 3), which is close to the 7.51 Tg derived from the experimental field measurements.

**4 Discussion**

**4.1 Contributions of different error sources to the uncertainties in the inventory**

Methane fluxes in rice paddies varied extensively with environmental and agronomic factors. Certain factors, such as rice biomass (Bachelet and Neue, 1993), organic matter input (Kern et al., 1995), water management (Khosa et al., 2011; Mishra et al., 1997), paddy soil properties (Yao et al., 1999; Gaunt et al., 1997), climate (Sass et al., 1991) and rice varieties (Ding et al., 1999), have previously been





recognized as having significant impacts on methane emissions. Other factors, such as atmospheric $CO_2$ and ozone contents (Bhatia et al., 2011; Inubushi et al., 2011), N fertilizer application (Banger et al., 2012; Xie et al., 2010a) and active soil organic C (Zhan et al., 2011), are also receiving increasing attention. Because so many factors affect the production, oxidation and emission of methane from rice cultivation, the

observed methane fluxes varied extensively both spatially and temporally. In the experimental field measurements (Fig. 1), the variations in rice paddy methane fluxes ranged from 3.2 kg $CH_4$ ha$^{-1}$ to 2,451.7 kg $CH_4$ ha$^{-1}$, averaging $341.2 \pm 383.2$ kg $CH_4$ ha$^{-1}$. The average simulated methane fluxes in the 10 x 10 km grids varied from 17.2 to 708.3 kg $CH_4$ ha$^{-1}$ (Fig. 6). The extremely high methane fluxes obtained from

experimental measurements were not reproduced by the model estimations. This discrepancy was partly due to the variations in the spatial representativeness of the methane fluxes in field observations and model estimations (Verburg et al., 2006).

The experimental measurements represented methane fluxes from an area of less than one hectare, while the modeled fluxes were the averages from 10 x 10 km grids.

This mismatch in spatial representativeness might also be due to errors in the model input data as well as to the impacts of other unknown factors (Singh and Dubey, 2012; Bhatia et al., 2011; Zheng et al., 2010; Gauci et al., 2008). Methane emissions could be estimated using a limited number of factors and simplified equations to express the complex relationships between methane emissions and influential factors, but such

simplification resulted in poor performance of the methods (Table 2). In Equation 5, $\sigma_{D,j}$ is the uncertainty due to errors in the input data. With an increasing number of explanatory factors, $B_R$ and $CV$ might decrease (which means better performance of the method), but $\sigma_{D,j}$ might increase because of the cumulative errors resulting from the increasing number of factors incorporated in the models. To reduce uncertainties

in the estimates and improve the performance of the model, the input data need to be available and of good quality. In the present study, model fallacy, represented by the equation $(F_j \times B_R)^2 + (F_j \times CV)^2$, contributed 79.5% – 88.9% to the uncertainty of $\sigma_{T,j}^2$, with $\sigma_{D,j}^2$ accounting for the remaining 11.1% – 20.5%. This implies that a model with better performance is needed to reduce the uncertainty of $\sigma_{T,j}$ in each grid.

The aggregated uncertainty of the national inventory depended not only on the magnitude of $\sigma_{CV,j}$ and $\sigma_{D,j}$ in each grid ($j$) but also on the spatial correlation between these variables (Equation C2 in the Supporting Information). The spatial correlation





of $\sigma_{D,j}$ depends on the availability of input data for the model and on spatial aggregation (Table C1 in the Supporting Information). However, the spatial correlation of $\sigma_{CV,j}$ could not be assessed analytically because it was a result of model imprecision, random noise and/or unknown factors. In the case of a strong correlation of $\sigma_{CV,j}$ values, the aggregated $\sigma_{CV}^2$ will account for a large proportion of $\sigma_T^2$ (right side in Fig. 8). However, if the spatial correlation is confined to a short distance, such as less than four grids (Dormann et al., 2007; Dray et al., 2006), the contribution of $\sigma_{CV}^2$ to $\sigma_T^2$ will be negligible (left side in Fig. 8). At the mid-point of $D_C$ (Equation C2, 30 grids, equal to 300 kilometers), as shown in Fig. 8, the model uncertainty ($\sigma_{BR}^2 + \sigma_{CV}^2$) accounted for 56.6% of the uncertainty in $\sigma_T^2$ (Table 4).

**4.2 Consistency of errors between model validation and model up-scaling**

Up-scaling a site-scale model (e.g., CH4MOD in this study) to a regional scale poses enormous challenges when data are scarce. Enhancing the spatial abundance of the input data minimizes the propagation of data error into the aggregated uncertainties. Many environmental and agricultural factors impact methane emissions from rice paddies. In the CH4MOD model, the key factors were parameterized as model inputs (Huang et al., 2004). However, when assessing the uncertainty of a model, the explanatory variables are arbitrarily included (Verburg et al., 2006). Li et al. (2004) found that soil properties were the "most sensitive factor" and therefore used this parameter in the uncertainty analysis. The inclusion of as many of the highly sensitive key factors as possible in the uncertainty analysis should generate more accurate and reliable results (right part in Fig. 2).

Experimental field studies have shown that the rice variety has substantial impacts on methane emissions (Aulakh et al., 2008; Inubushi et al., 2011; Jia et al., 2002). A study of field observations (Su et al., 2015) showed that transfer of the barley gene *SUSIBA2* to rice favors the allocation of photosynthesis to the aboveground biomass over allocation to the roots and, moreover, that less biomass allocation to root exudates results in reduced methane emissions. The impact that the rice variety has on methane emissions was parameterized as the variety index (*VI*) in CH4MOD. According to Huang et al. (1998), *VI* ranges from 0.5 to 1.5 and averages 1.0 for most rice varieties. To validate the CH4MOD model (left portion of Fig. 2) using the 495 methane emission measurements included in the present study, *VI* was assigned a default value of 1.0 regardless of the rice variety because, until now, no



dedicated attempts have been made to quantify the *VI* of different rice varieties. Therefore, the *BR* and *CV* values presented in Table 2 incorporate the uncertainty in model performance that can be attributed to different rice varieties ($M_f(x)$ in Equation C2 of the Supporting Information). To maintain consistency, *VI* was assigned the same default value (1.0) when the model was scaled-up to the national scale (right

side of Fig. 2), and no PDF was built for the uncertainty calculation conducted with the Monte Carlo simulation. If a PDF had been incorporated into the uncertainty calculation when the model was scaled-up, the overall uncertainties (Table 4) would have been overestimated. However, if different *VI* values were assigned to rice varieties during model validation, the error caused by the inaccuracy of *VI* would also

need to be considered during the scaling-up of the model to prevent underestimation of the overall uncertainty.

**5 Conclusions**

    Due to the remarkable spatial variation in rice paddy methane emissions, the uncertainties in regional estimates obtained either through field measurements or

modeling remain considerably large. For field measurements, the reduction in uncertainty achieved by increasing the number of observations was shown to be inversely related to the spatial correlation between the measurements. To reduce the estimation bias, the number of measured emission fluxes should be proportional to the paddy area where the corresponding agronomic activities and environmental

conditions occur homogenously.

    Model performance depends not only on the effectiveness of the models themselves but also on the availability of the data needed to drive the model. We found that without a sufficient quantity of high-quality data, a well-developed model will perform even more poorly than simple regression approaches. When modeling

methane emissions, uncertainties in the performance of the model remain the major obstacle to reliably estimating methane emissions. Estimate uncertainty could be reduced at the regional scale by increasing the availability of input data and decreasing spatial correlations among the residues of the model output.





**Appendix A** Area-weighted methane flux measurements

For the area-weighted analysis, the mean ($m_w$) and variances ($V_w$) of the methane flux measurements ($O_i$) were calculated as:

$$m_w = \sum_{i=1}^{N} w_i \times O_i \tag{A1}$$

$$V_w = \sum_{i=1}^{N} w_i \times (O_i - m_w)^2 \tag{A2}$$

constrained by $\sum_{i=1}^{N} w_i = 1$, where $N$ is the total number of all methane flux measurements. Ordinarily, $w_i = \dfrac{1}{N}$. However, here, we scaled $w_i$ by the area proportional to that of each water regime applied in rice cultivation in China:

$$w_{wr} = \frac{R_{wr}}{N_{wr}}, \; wr \in \{continuously\ flooding,\ single\ drainage,\ multi-drainage\} \tag{A3}$$

where $N_{wr}$ is the number of measurements belonging to water regime $wr$, and $R_{wr}$ is the proportion of the area of rice paddies irrigated with each of the three water regimes. $R_{wr}$ assumes values of 0.1, 0.2 and 0.7, respectively, for the three water regimes according to previous research (Mao, 1981; Xiong et al., 1992; Li, 2002; Zou et al., 2009). If a methane flux measurement ($i$) in Equations A1 and A2 belongs to water regime $wr$, then $w_i = w_{wr}$.

The standard error (SE) of the area-weighted mean ($m_w$) is calculated as follows:

$$SE = \sqrt{\frac{V_w}{N-1}} \tag{A4}$$

It should be noted that Equation A4 only holds when the measurements are statistically independent; if this is not the case, mostly due to spatial correlations of the environmental conditions that support the measurements, then the value for $N$ should be smaller, depending on the strength of the correlation (Bence, 1995).

**Appendix B** CH4MOD model and the datasets used for simulating national rice paddy methane emissions

CH4MOD is an empirical model that simulates methane production and emissions from rice paddies under various environmental conditions and agricultural practices (Huang et al., 1998, 2004; Xie et al., 2010a). It calculates methanogenic substrate production from rice plant root exudates and added organic matter (OM) decomposition. Both OM decomposition and rice plant-induced substrate production are significantly influenced by environmental factors, including the soil texture and temperature, with the soil moisture content controlling the fraction of transformation



of the substrates into methane. There are two major routes by which methane produced in rice paddy soils escapes into the atmosphere: via the arenchyma system of the rice plants and via methane bubbles. Both of these pathways are incorporated into the model.

CH4MOD runs in a daily step, driven by the daily air temperature. Its input parameters include the soil sand percentage (*SAND*), organic matter amendment (*OM*), rice grain yield (*GY*), water management pattern ($W_{ptn}$) and rice cultivar index (*VI*).

*Rice harvest area and grain production*

Data on rice production and the harvest area for each province in 2005 were obtained from China's Statistical Yearbook (EBCAY, 2006) for early, late and middle rice. County-level rice production data were obtained from censusing conducted by the Chinese Academy of Agricultural Sciences. Although the fractions of early, late and single rice cultivation are not included in the county-level data, the rotation type for each county was formulated using the approach of Frolking et al. (2002) by referring to the climatic zones of each cropping system in China (Han et al., 1987).

Several studies have shown that methane emissions differ significantly among rice varieties (Singh et al., 1997; Wang et al., 1999). In CH4MOD, the impact of the methane variety on methane emissions was parameterized as the variety index (*VI*) (Huang et al., 1998, 2004). *VI* ranges from 0.5 to 1.5 but is typically approximately 1.0 for most rice varieties (Huang et al., 1997, 2004).

*Climate data and rice phenologies*

The daily mean air temperature is the only meteorological data required to drive the CH4MOD model. Air temperature data were obtained for 2005 from 678 meteorological stations included in the National Meteorological Information Center (NMIC), China Meteorological Administration (CMA) (http://cdc.cma.gov.cn/) database. For counties that lack a meteorological station, air temperature data from the nearest neighbouring station were substituted.

Rice phenologies (specifically transplanting and harvesting dates) control the start and end of the CH4MOD run for simulating methane emissions. Data regarding rice phenologies were originally derived from iso-line maps edited by Zhang et al. (1987) in the Agricultural Climate Atlas of China. The transplanting and harvesting dates within each grid were spatially interpolated from the iso-lines via the TIN (triangular irregular network) technique (Aumann et al., 1991) and assigned to each county.

*Soil properties*

The spatial database for the soil sand content (*SAND*) is part of the databases developed by the Institute of Soil Sciences, Chinese Academy of Sciences, from the samples of soil profiles obtained during the Program of the Second Soil Survey of





China and subsequent surveys. The database comprises of 10 km × 10 km raster
datasets of soil properties at 10-cm intervals from the surface down through the
profile, making the spatial resolution of soil the finest of the CH4MOD input
parameters.

*Organic matter amendment and water regimes in rice paddies*
    The organic matter inputs into rice fields include various types of farm manure
(e.g., green manure, animal feces) and crop straw as well as dead roots and stubble
from previous crops. Root biomass remaining in the soil can be calculated using the
root/shoot ratio (Huang et al., 2007). Stubble biomass was assumed to be one-tenth
the aboveground straw biomass. However, the fractions of straw incorporation and
farm manure application are not well known, and the data are therefore limited. In the
First National Census of Pollution Sources conducted by the Ministry of
Environmental Protection of China (CFPC, 2011), straw application in croplands was
summarized at the provincial level in the census data (Table B1); thus, the value for
straw application given in Table B1 is not rice specific but accounts for all crops in
each province. This bias may not be significant in provinces where crop cultivation is
dominated by rice. In addition to crop straw, the incorporated crop residues include
dead crop roots and stubble; according to Zhao and Li (2001), stubble accounts for
approximately 13% of the total dry weight of straw.
No regular statistical data or comprehensive census data were available for manure
application in rice cultivation. In this study, we estimated OM application in rice
cultivation by examining more than 1000 research papers; estimates of farmyard
manure application in each province are shown in Table B1.
    Since the mid-1960s, a diverse array of irrigation regimes have been adopted that
diverge from the traditional approach of continuous flooding, representing an
important development for rice cultivation in China (Xiong et al., 1992; Li, 2002;
Peng et al., 2007). As such, different compositions of flooding, drainage and moisture
irrigation have been applied according to the climate, soil and topographic conditions
of the rice fields and factors such as the rice variety being grown, its developmental
stage and hydrological construction. To simplify CH4MOD, the forms of irrigation
used for rice cultivation were grouped into five general irrigation patterns: 1)
flooding-drainage-flooding-intermittent irrigation, 2) flooding-drainage-intermittent
irrigation, 3) flooding-intermittent irrigation, 4) continuous flooding and 5)
continuously intermittent irrigation (Gao and Li, 1992; Huang et al., 2004). Despite
being the agronomic factor that is most sensitive to methane emissions (Table B2), the
available data on irrigation are the scarcest among all of the inputs needed for
CH4MOD up-scaling. Except for a few brief mentions in the literature (Mao, 1981;
Xiong et al., 1992; MWRUC, 1996; Cai, 2000; Ma et al., 2005), almost no detailed
data addressing spatial variations in rice irrigation are available. Given this limitation,



we made rough assumptions about irrigation for each grand region of rice cultivation
(Fig. 1, Table B2).

**Appendix C**  Uncertainties in regional estimates obtained via the modeling approach

$F(x)$ is a spatial process that has a determinative component, $D(x)$, and a random
component, $R_0(x)$:

$$F(x) = D(x) + R_0(x) \qquad (C1)$$

where $x$ is the location of two dimensions in the spatial domain, $S$.

When $D(x)$ is implemented in a model $M(x)$ that simulates the spatial variation of
$D(x)$, there is unavoidably an error component, $M_f(x)$ (the model fallacy), due to the
imperfection of the model, and therefore

$$D(x) = M(x) + M_f(x) \qquad (C2)$$

Combining $M_f(x)$ and $R_0(x)$ into one component, $R_m(x)$, $F(x)$ this expression can be
rewritten as

$$F(x) = M(x) + R_m(x) \qquad (C3)$$

where $R_m(x)$ is typically used to evaluate model performance. To explicitly address
the model input variables, e.g., environmental factors and anthropogenic activities of
the model mechanism, $M(x)$ can be expressed as

$$F(x) = M(v_1, v_2, v_3, \ldots) + R_m(x) \qquad (C4)$$

where $v_1$, $v_2$, $v_3$, … are the model input variables. Averaging over the $S$ domain,
Equation C4 yields:

$$\overline{F(x)} = \overline{M(v_1, v_2, v_3, \ldots)} + \overline{R_m(x)} \qquad (C5)$$

To implement the averaging of the model simulation over the spatial domain, the
theoretical approach is

$$\overline{M(v_1, v_2, v_3, \ldots)} = \frac{\int_S M(v_1, v_2, v_3, \ldots)dx}{S} \qquad (C6)$$

However, because it is impossible to obtain data for the model input variables at
every location, $x$, of domain $S$, $\overline{M(v_1, v_2, v_3, \ldots)}$ has to be represented by the model
simulation $M(v_1^{(p)}, v_2^{(p)}, v_3^{(p)}, \ldots)$ at a specific location, $p$, and there emerges the
representative error, $R_s$, which conforms to

$$\overline{M(v_1, v_2, v_3, \ldots)} = M(v_1^{(p)}, v_2^{(p)}, v_3^{(p)}, \ldots) + R_s \qquad (C7)$$

The representative error, $R_s$, comes from both the imprecision and poor spatial
availability of the model inputs. The magnitude of $R_s$ therefore depends on the model
input errors and how sensitive the model simulation is in response to the variation of
the model inputs. Combining Equations C5 and C7 gives us

$$\overline{F(x)} - M(v_1^{(p)}, v_2^{(p)}, v_3^{(p)}, \ldots) = R_s + \overline{R_m(x)} \qquad (C8)$$





The right side of Equation C8 is therefore the error of the model simulation over the spatial domain. From its definition in Equation C7, the statistical expectation of $R_s$ is 0, and the uncertainty of $R_s$ (the sum of the squared $R_s$ occurrence) is equivalent to its statistical variance. By referring to the structural analysis of the model residue error, $\overline{R_m(x)}$, given in Allen and Raktoe (1981), we itemized the total uncertainty ($U_T$)

of the model simulation over the spatial domain as

$$U_T = U_D + U_{BR} + U_{CV} \tag{C9}$$

where $U_D$ signifies the spatially representative error corresponding to $R_s$ in Equation C8, and $U_{BR}+U_{CV}$ is the uncertainty attributed to the model fallacy, $R_m(x)$. $U_{BR}$ represents the model performance bias at the site scale, whereas $U_{CV}$ represents

the model fallacy error apart from $U_{BR}$, which is the combination of the regression error and the random error, as described in Allen & Raktoe (1981). The assumed independence between $R_s$ and $R_m(x)$ originates from the fact that they are due to separate causes. For a specific model, the model fallacy is independent of the accuracy and availability of the model inputs that facilitate modeling in a spatial

domain. However, changes in the model mechanism may regulate the relationship between $U_D$ and $U_{BR}+U_{CV}$; for example, improving model performance by incorporating more factors as input variables may reduce the model's fallacy but increase the representative error due to the additional input data requirements necessary to run the model .

Due to substantial heterogeneities in spatial processes, such as fluxes in methane emissions from rice cultivation, the large area under study is usually split into several smaller regions. These regions may consist of grids of the same size or irregular patches of different sizes. Each division is a spatial domain with less heterogeneity to which modeling can be applied. To summarize the modeling results for each division,

the spatial aggregation of $U_D$ was discussed by Zhang et al. (2014) and is briefly addressed in Appendix D. Appendix D also provides the rationale for the spatial aggregation of $U_{BR}$ and $U_{CV}$.

**Appendix D** Spatial aggregation of the estimation uncertainties in grids
**D.1** Correlation coefficients of the model estimates between two grids due to data

sharing of the model inputs

In each grid, $i$, the model estimates obtained via Monte Carlo iteration produce a numeric depiction of a random variable $V_i(m_i, \sigma_i)$, where $m_i$ and $\sigma_i$ are the statistical mean and standard deviation, respectively, of the random variable $V_i$. Thereafter, model up-scaling involves summation of the random variables $V_0=V_1+V_2+...+V_N$. The

aggregation of uncertainty, represented by the statistical variance or standard deviation, is generalized as $Var(\sum_{i=1}^{N} X_i) = \sum_{i=1}^{N}\sum_{j=1}^{N} Cov(X_i, X_j)$ (Ross, 2006), and it can



be transformed into a quadratic summation of the elementary variances via the standardized variance–covariance matrix:

$$\sigma_0^2 = \sum_{i,j} \sigma_i \times C_{ij} \times \sigma_j \; , \; (i=1...N, j=1...N) \tag{D1}$$

where $\sigma_0^2$ is the aggregated variance of the regional estimation, and $\sigma_i$ and $\sigma_j$ are the standard deviations of the within-cell variations in cells $i$ and $j$, respectively.

Matrix $\mathbf{C}$ is composed of $C_{ij}$ coefficients, which represent "correlations" between the spatially representative errors ($R_s$ in Equation C8) of the individual cells. "Correlation" here is a measure of how the model outputs in two cells vary concurrently when they share common data for the model inputs. If the estimate in cell $i$ is over-/underestimated, then the estimate in cell $j$ will most likely be over-/underestimated as

well, and vice versa, because they share common data. It is noteworthy that the correlation represented by $C_{ij}$ is different from that between the "real" processes represented by $F(x)$ in Equation C8. The aggregation of the model outputs can be quite simple if the model estimate is generated with independent data in each cell. In this case, matrix $\mathbf{C}$ will be an identity matrix in which the diagonal elements will be 1,

and all of the off-diagonal elements will be 0. The aggregation in Equation D1 will thereafter indicate the arithmetic sum of the within-cell variances, as addressed by the *Law of Large Numbers*. However, when there are not sufficient data to support independent calculation among cells, the off-diagonal elements, $C_{ij}$, of matrix $\mathbf{C}$ will no longer be zero.

In the present study, $C_{ij}$ was empirically calculated through numerical experiments. For a different level of data sharing between two cells (Table D1), the model estimates in the two cells were iteratively calculated with CH4MOD. The model inputs were randomly selected from the range of values for the variables. When data sharing occurred between the two cells for a variable in Table D1, the value of the

variable was selected once for the two cells; for the variables for which there was no data sharing, the value of the variable was selected separately for the two cells. The correlation coefficients ($C_{ij}$) of the model estimates in the two cells were statistically calculated with 1000 iterations of the paired model estimates in the two cells in the present study.

**D.2** Spatial aggregation of estimation uncertainties

Equation D2, an alternative form of Equation C9 in Appendix C, is used to aggregate the uncertainties in all grids to calculate the uncertainty in the national inventory:





$$\sigma_T{}^2 = \sigma_{BR}^2 + \sigma_{CV}^2 + \sigma_D^2$$

$$= \sum_{j=1}^{N} \sum_{k=1}^{N} (A_j \times F_j \times B_R) \times E_{j,k} \times (A_k \times F_k \times B_R)$$

$$+ \sum_{j=1}^{N} \sum_{k=1}^{N} (A_j \times F_j \times CV) \times U_{j,k} \times (A_k \times F_k \times CV) \qquad \text{(D2)}$$

$$+ \sum_{j=1}^{N} \sum_{k=1}^{N} (A_j \times \sigma_{D,j}) \times D_{j,k} \times (A_k \times \sigma_{D,k})$$

In Equation D2, $A_j$ and $A_k$ represent the rice harvesting area in grids $j$ and $k$, respectively. No errors in the rice harvesting area were considered in Equation D2. $F_j$ and $F_k$ represent the average methane fluxes from the Monte Carlo simulation in grid $j$ and $k$, and $\sigma_{D,j}$ and $\sigma_{D,k}$ are the standard deviations associated with $F_j$ and $F_k$, respectively. No cross-correlation between the three components was considered here. Because of limited data availability, the neighboring grids were assigned probabilities of sharing data for the model input variables. The aggregation of $\sigma_{D,j}$ in the grids was therefore kernelled using data-sharing matrix $\mathbf{D}$ ($D_{j,\,k}$ represents its element, Table D1). $E_{j,\,k} = 1$ is the element of a constant matrix, $\mathbf{E}$, which refers to the bias of the model estimates in all grids and is statistically under/overestimated concurrently in all grids. $U_{j,\,k}$ is the element of matrix $\mathbf{U}$. $U_{j,\,k}$ is not specifically known. The two extremes of matrix $\mathbf{U}$ correspond to matrix $\mathbf{E}$ and the identity matrix, $\mathbf{I}$. The estimation error, $F_j \times CV$, is related to the factors that are not explicitly accounted for in the model, for instance, mineral fertilizer application (Xie et al., 2010a) and soil organic carbon content (Zhan et al., 2011). Because the inter-grid relationships of these "unknown" factors could not be explicitly accounted for, we assigned $\mathbf{U}$ the values for the mid-point of the two extremes $\mathbf{E}$ and $\mathbf{I}$.

### Acknowledgements

The study was jointly supported by the National Natural Science Foundation of China (Grant No. 41573069, 31370492, 41321064). We would also thank the Resources and Environmental Scientific Data Centre of the Chinese Academy of Sciences and the National Meteorological Information Center of the Chinese Meteorological Administration for their support in providing the data.



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





5   Tables

**Table 1 Methods and their input scenarios**

| Methods | Input scenario | Reference |
|---|---|---|
| R1:  $C_{CH4} = 0.3 \times C_{input}$ | R1－S0: Case-specific C input, adjusted with the water regime. | Neue et al., (1990) |
| R2:  $CH_4 = -0.006 \times C_{input} + 0.078 \times N_{input} + 0.885 \times R_{C/N} + 21.15$ | R2－S0: Case-specific C and N input.<br>R2－S1: Case-specific C input, averaged N input in all cases. | Kern et al., (1997) |
| M: CH4MOD model | M－S0: Case-specific inputs of all model variables: e.g., organic matter amendments, soil properties and water regimes | Huang et al., (1998, 2004); Xie et al., (2010a) |
| | M－S1: Case-specific inputs of soil properties and water regimes; other model variables use averaged values for all 495 cases | |
| | M－S2: Case-specific inputs of organic matter amendments; all other model variables use averaged values for all 495 cases, the water regime was assumed to be multi-drainage irrigation. | |
| | M－S3: No case-specific inputs used for soil properties or organic matter amendments, the water regime was assumed to be multi-drainage irrigation. | |

† Regression equation R1 was developed according to measurements conducted in continuously flooded fields, and the calculated flux was therefore adjusted by a scaling factor of 1.00, 0.65 or 0.56 for continuous flooding, single drainage or multi-drainage irrigation, respectively (IPCC, 2006).

10   ‡ The water regimes in the CH4MOD model (Huang et al., 2004) are more specifically defined and differ from that of the IPCC (2006).



**Table 2 Performance of the methods under different scenarios of data availability**

| Method | Bias of the estimation ($B_R$) | Std. of the estimation residues ($CV$) | Total error[‡] |
|---|---|---|---|
| R1－S0 | 212.0 (62.1%)[†] | 577.1 (163.3%) | 596.0 (174.7%) |
| R2－S0 | −1.3 (−0.4%) | 407.8 (119.5%) | 407.8 (119.5%) |
| R2－S1 | −4.9 (−1.4%) | 415.7 (121.8%) | 415.7 (121.9%) |
| M－S0 | −24.2 (−7.1%) | 251.8 (73.8%) | 253.0 (74.1%) |
| M－S1 | −30.8 (−9.0%) | 343.9 (100.8%) | 345.2 (101.2%) |
| M－S2 | −120.7 (−35.4%) | 341.3 (100.0%) | 362.9 (106.1%) |
| M－S3 | −109.8 (−32.2%) | 401.8 (117.8%) | 416.6 (122.1%) |

† Percentages in parentheses indicate the magnitude of the error relative to the overall

average methane flux ($m_c$) for all cases, and $m_c$ = 341.2 kg CH$_4$ ha$^{-1}$ (Fig. 2a).

‡ *Total error* $= \sqrt{B_R^2 + CV^2} \times m_c$



**Table 3 Methane emissions inventory and the uncertainties caused by model imperfection and errors in model input data**

| Rice | Harvesting area ($\times 10^6$ ha) | CH$_4$ emission (Tg) | $\sigma_T$(Tg)[‡] | 95% CI[§] (Tg) |
|---|---|---|---|---|
| Early rice | 5.96 | 1.22 | 0.39 | 0.58 — 2.08 |
| Late rice | 5.96 | 1.52 | 0.40 | 0.85 — 2.39 |
| Single rice | 16.93 | 3.69 | 0.75 | 2.37 — 5.30 |
| All rice | 28.85 | 6.43 | 1.53 | 3.79 — 9.77 |

[‡] Calculated with Equation 6;

[§] 95% CIs were calculated by assuming the gamma probability distributions, for which the shape and scale parameters were estimated via momentum methods.





**Table 4 Components of the uncertainty in the national inventory**

| Rice | Due to model performance | | Due to data quality and availability, $\sigma_D{}^2$ | Total | |
|---|---|---|---|---|---|
| | $\sigma_{BR}{}^2$ | $\sigma_{CV}{}^2$ | | $\sigma_T{}^2$ | $\sigma_T$ |
| Early rice | 0.01 | 0.06(0.00－0.81)[‡] | 0.08 | 0.15 | 0.39 |
| Late rice | 0.01 | 0.10(0.00－1.28) | 0.05 | 0.16 | 0.40 |
| Single rice | 0.07 | 0.25(0.00－5.15) | 0.24 | 0.56 | 0.75 |
| All rice | 0.21 | 1.12(0.00－22.56) | 1.00 | 2.35 | 1.53 |

[‡] Numbers in parentheses represent the range of $\sigma_{CV}{}^2$ depending on the spatial correlation of the model simulation residuals. Long-distance correlation results in a large aggregated $\sigma_{CV}{}^2$ value, whereas short-distance correlation results in a small aggregated $\sigma_{CV}{}^2$ value.



**Table B1 Fraction of straw incorporation and farm manure application in rice cultivation [£]**

| Province | Fraction of straw incorporation[†] | Farm manure (kg OM ha$^{-1}$)[‡] | | Province | Fraction of straw incorporation | Farm manure (kg OM ha$^{-1}$) | |
|---|---|---|---|---|---|---|---|
| | | Mean | Range | | | Mean | Range |
| Beijing | 0.41 | 821.6 | 321.6－1321.6 | Henan | 0.56 | 1539.2 | 776.2－2302.1 |
| Tianjin | 0.29 | 927.4 | 123.1－1731.6 | Hubei | 0.20 | 2101.3 | 981.1－3221.6 |
| Hebei | 0.62 | 1519.3 | 959.5－2079.1 | Hunan | 0.34 | 1836.9 | 846.7－2827.2 |
| Shanxi | 0.44 | 1824.8 | 1195.5－2454.2 | Guangdong | 0.23 | 1243.2 | 634.5－1851.8 |
| Inner Mon. | 0.12 | 1837.5 | 1042.4－2632.7 | Guangxi | 0.27 | 1384.7 | 645.4－2124.1 |
| Liaoning | 0.03 | 1108.5 | 657.8－1559.3 | Hainan | 0.22 | 1408.5 | 964.8－1852.1 |
| Jilin | 0.03 | 1308.4 | 421.5－2195.4 | Chongqing | 0.17 | 1608.7 | 801.5－2415.8 |
| Heilongjiang | 0.23 | 1800.8 | 836.0－2765.6 | Sichuan | 0.18 | 1922.7 | 940.7－2904.7 |
| Jiangsu | 0.23 | 1263.5 | 605.6－1921.4 | Guizhou | 0.09 | 1793.2 | 740.2－2546.1 |
| Zhejiang | 0.35 | 1276.2 | 734.1－1818.3 | Yunnan | 0.10 | 1802.3 | 853.1－2751.5 |
| Anhui | 0.19 | 1507.5 | 424.3－2590.7 | Shaanxi | 0.34 | 1769.6 | 555.3－2983.9 |
| Fujian | 0.32 | 1123.1 | 852.6－1393.6 | Gansu | 0.03 | 1923.0 | 375.9－3470.1 |
| Jiangxi | 0.38 | 1612.2 | 842.3－2382.1 | Ningxia | 0.15 | 1448.6 | 515.5－2381.7 |
| Shandong | 0.55 | 1032.8 | 530.8－1534.7 | Xinjiang | 0.45 | 1612.0 | 407.7－2816.3 |

£ No data on farm manure application were available for Shanghai and Tibet; as such, data for Jiangsu and Guizhou, respectively, were used as substitutes.

† Statistics derived from the First National Pollution Source Census conducted by the Ministry of Environmental Protection of China (CFPC, 2011); however, the range of variation was not provided in the publication.

‡ Statistics derived from an investigation of organic application in crop cultivation performed by the Institute of Atmospheric Physics, Chinese Academy of Sciences. Green manure was not included because it accounts for a minor proportion of the total organic matter application in rice cultivation.





5    **Table B2 Proportions of different water irrigation patterns[†] in each grand region**

| Grand region[‡] | Baseline fraction | Uncertainty fraction |
|---|---|---|
| I | 3: 0.92; 4: 0.08[£] | 1: 0.31; 2: 0.31; 3: 0.30; 4: 0.08 |
| II | 2: 0.95; 4: 0.05 | 1: 0.32; 2: 0.32; 3: 0.31; 4: 0.05 |
| III | 2: 0.82; 4: 0.18 | 1: 0.27; 2: 0.28; 3: 0.27; 4: 0.18 |
| IV | 1: 1.0 | 1: 0.34; 2: 0.33; 3: 0.33 |
| V | 1: 1.0 | 1: 0.34; 2: 0.33; 3: 0.33 |

† Refer to Huang et al. (2004) for the definition of water irrigation patterns

‡ Grand region I: Guangdong, Guangxi, Hainan, Hunan and Jiangxi; Grand region II: Fujian, Hubei, Zhejiang, Jiangsu, Shanghai and Anhui; Grand region III: Chongqing, Sichuan, Yunnan and Guizhou; Grand region IV: Heilongjiang, Liaoning and Jilin; Grand region V: other provinces.

10  £ Indicates that water irrigation pattern 3 was applied in 92% of the rice cultivation area in Grand region I (Fig. 2a), and the remaining 8% of the rice area was under continuous flooding (water irrigation pattern 4).



**Table D1 Look-up table of correlation coefficients of the model outputs in two cells due to data sharing**

| Yield | OM | Sand | $W_{Ptn}$ | VI | $C_{ij}$ | Yield | OM | Sand | $W_{Ptn}$ | VI | $C_{ij}$ |
|---|---|---|---|---|---|---|---|---|---|---|---|
| 0† | 0 | 0 | 0 | 1 | 0.069 | 1 | 0 | 0 | 0 | 1 | 0.136 |
| 0 | 0 | 0 | 1 | 0 | 0.347 | 1 | 0 | 0 | 1 | 0 | 0.430 |
| 0 | 0 | 0 | 1 | 1 | 0.413 | 1 | 0 | 0 | 1 | 1 | 0.520 |
| 0 | 0 | 1 | 0 | 0 | 0.295 | 1 | 0 | 1 | 0 | 0 | 0.343 |
| 0 | 0 | 1 | 0 | 1 | 0.375 | 1 | 0 | 1 | 0 | 1 | 0.478 |
| 0 | 0 | 1 | 1 | 0 | 0.674 | 1 | 0 | 1 | 1 | 0 | 0.776 |
| 0 | 0 | 1 | 1 | 1 | 0.796 | 1 | 0 | 1 | 1 | 1 | 0.900 |
| 0 | 1 | 0 | 0 | 0 | 0.082 | 1 | 1 | 0 | 0 | 0 | 0.170 |
| 0 | 1 | 0 | 0 | 1 | 0.167 | 1 | 1 | 0 | 0 | 1 | 0.225 |
| 0 | 1 | 0 | 1 | 0 | 0.436 | 1 | 1 | 0 | 1 | 0 | 0.481 |
| 0 | 1 | 0 | 1 | 1 | 0.519 | 1 | 1 | 0 | 1 | 1 | 0.616 |
| 0 | 1 | 1 | 0 | 0 | 0.396 | 1 | 1 | 1 | 0 | 0 | 0.458 |
| 0 | 1 | 1 | 0 | 1 | 0.499 | 1 | 1 | 1 | 0 | 1 | 0.575 |
| 0 | 1 | 1 | 1 | 0 | 0.760 | 1 | 1 | 1 | 1 | 0 | 0.849 |
| 0 | 1 | 1 | 1 | 1 | 0.878 | 1 | 1 | 1 | 1 | 1 | 1.000 |
| 1 | 0 | 0 | 0 | 0 | 0.066 | | | | | | |

†1 means that the two cells share data for the variable, and 0 means that they do not share data for the variable



## 5 Figure Legends

Figure 1 Locations of the experimental sites (red stars). The background map is the spatial distribution of rice paddy in China. The size of the red star is proportional to the number of the measured methane fluxes at the site. The polygons show zones of different crop

rotation systems involving rice: I—Double rice rotation, II—Mixing zone of rice/rice rotation and rice/upland crop rotation, III & IV—Rice/upland crop rotation or rice/fallow rotation, V & VI—Rice/fallow rotation, and VII—No rice area.

Figure 2 Overall flowchart in estimating regional/national methane emissions and the uncertainties

by field measurements and modeling

Figure 3 Statistical description of the measured methane fluxes. (a) Statistical parameters, and (b) Histogram of the measurements. The solid circles represent the sample mean and the vertical bars are 95% confidence intervals of the samples, from the 2.5% percentile to the

97.5% percentile. The dashed line indicates the arithmetic average of all measured fluxes ($m_c$). The solid line is the area-weighted mean of the methane fluxes ($m_w$), by referencing to the areal proportion of each water regime in the national total rice harvesting area, 10% continuously flooding, 20% single-drainage and 70% multi-drainage (Zou *et al*., 2009; MWRUC, 1996; Li *et al.,* 2001; Xiong *et al.,* 1992).

Figure 4 P-P plots of the Cumulative probability of the measured methane fluxes vs. the Gamma distribution. (a) Single drainage irrigation cases, (b) multi-drainage irrigation cases, (c) continuously flooding irrigation cases, and (d) all cases after being area-weighted (Appendix A). The *n*, *avg.* and *std.* is the sample size, statistical mean and standard

deviation of the sample methane fluxes, respectively. The *α* and *β* is the shape and scale parameters of the Gamma distribution that were calculated with the statistical mean and variance of the measured methane fluxes, $\beta = (std.)^2/(avg.)$ and $\alpha = (avg.)/\beta$. The diagonal line is the 1:1 straight line for a perfect Gamma distribution match.



5    Figure 5 The measured methane fluxes in the experiments against their simulation results by
different methods. (a) R1－S0, (b) R2－S0 and (c) M－S0 which are described in Table 1.

Figure 6 Histograms and their fitting Gamma probability lines of the calculated methane fluxes of
the 10 km by 10 km rice paddy grids of China. (a) single rice, including rice-fallow and
10    rotations of rice with upland crops; (b) and (c) are early and late rice in double rice
rotations. The vertical bars are the histograms of the calculated $F_j$ (Equation 5), and the
solid line is the theoretic Gamma PDF line with parameters derived from statistics of $F_j$
via momentum methods.

Figure 7 Spatial distributions of rice paddy methane emissions ($\times 10^6$ g $CH_4$ per 10km$\times$10km grid).

Figure 8 Composition of the aggregated uncertainty of the national inventory along with the
spatial autocorrelation in variances of the modeling residues in grids. The distance criteria
($D_c$) is used to define a step function of spatial autocorrelation: if two grids depart by a
distance beyond $D_c$, the autocorrelation is 0, otherwise, it is 1. The step function is a
20    simplified and the upper limit of the true spatial autocorrelation. With the step function,
larger $D_c$ indicates stronger autocorrelation.





Figure 1

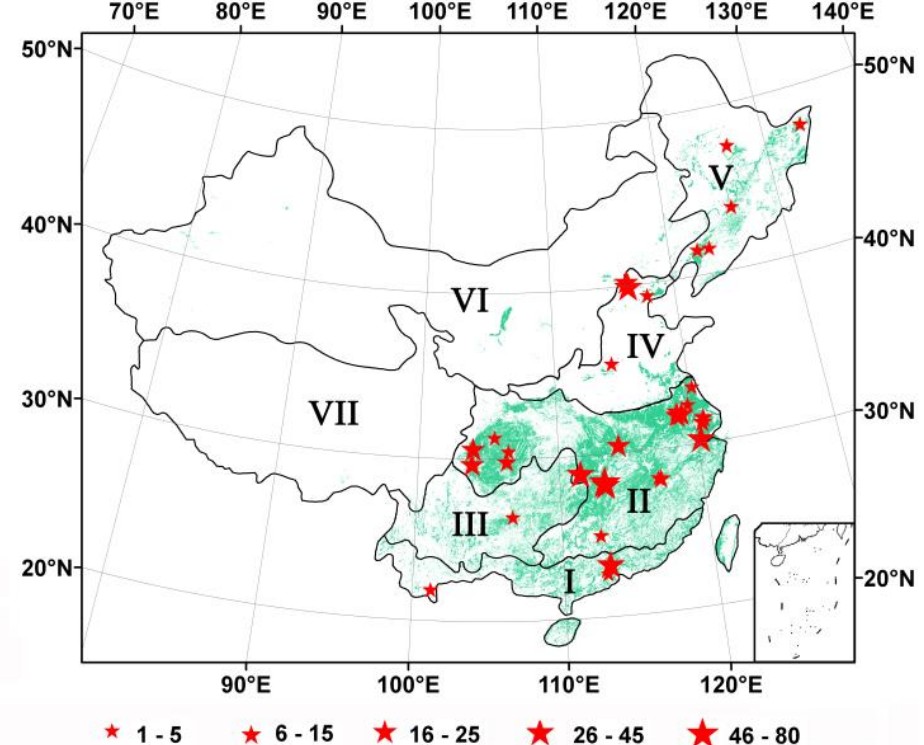

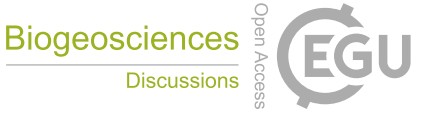

Figure 2

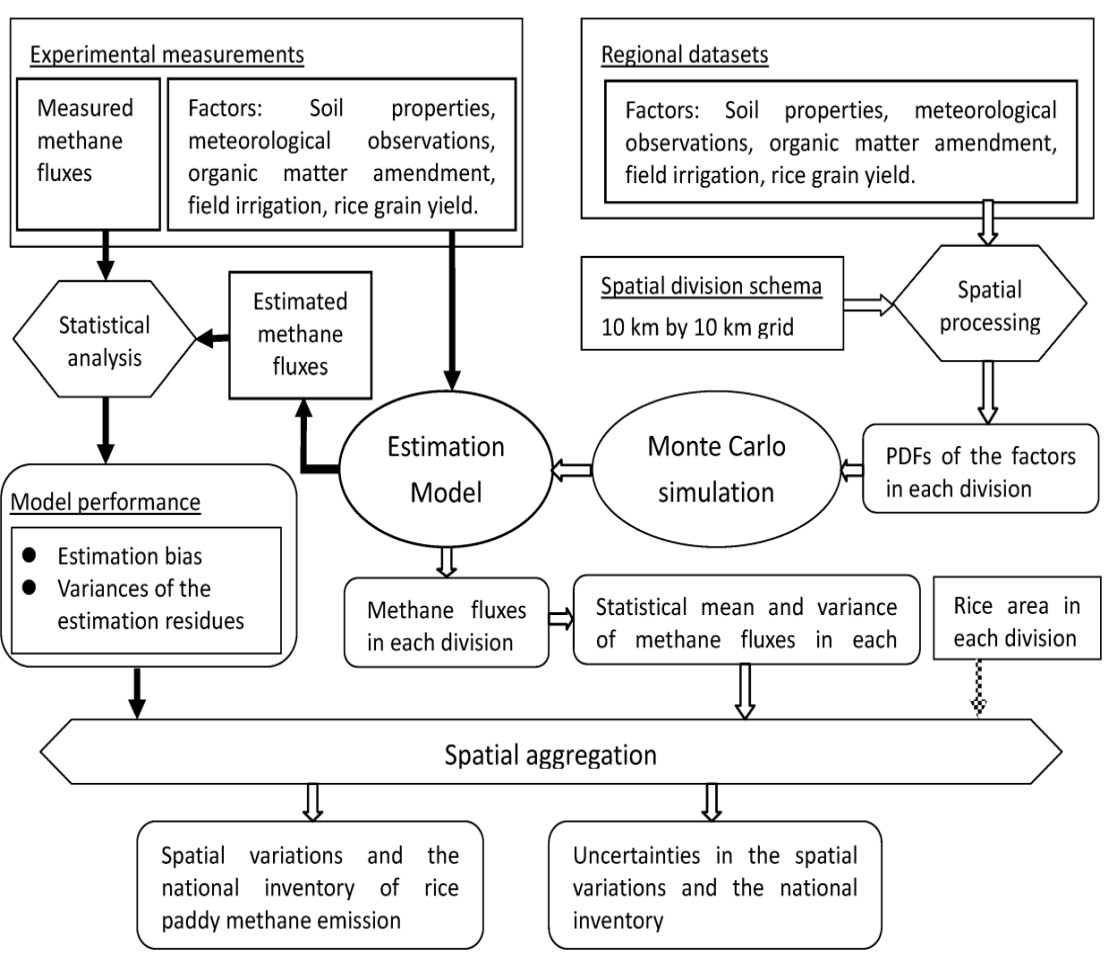





5      Figure 3

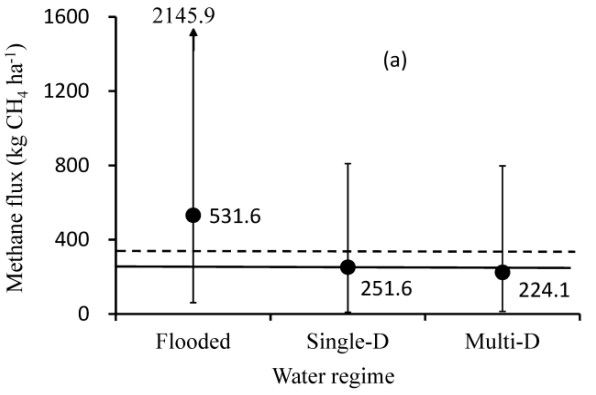
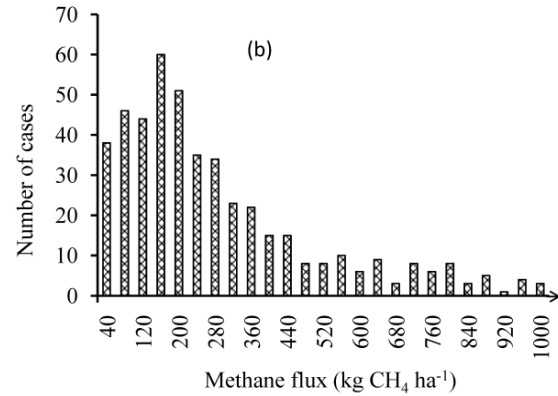




Figure 4

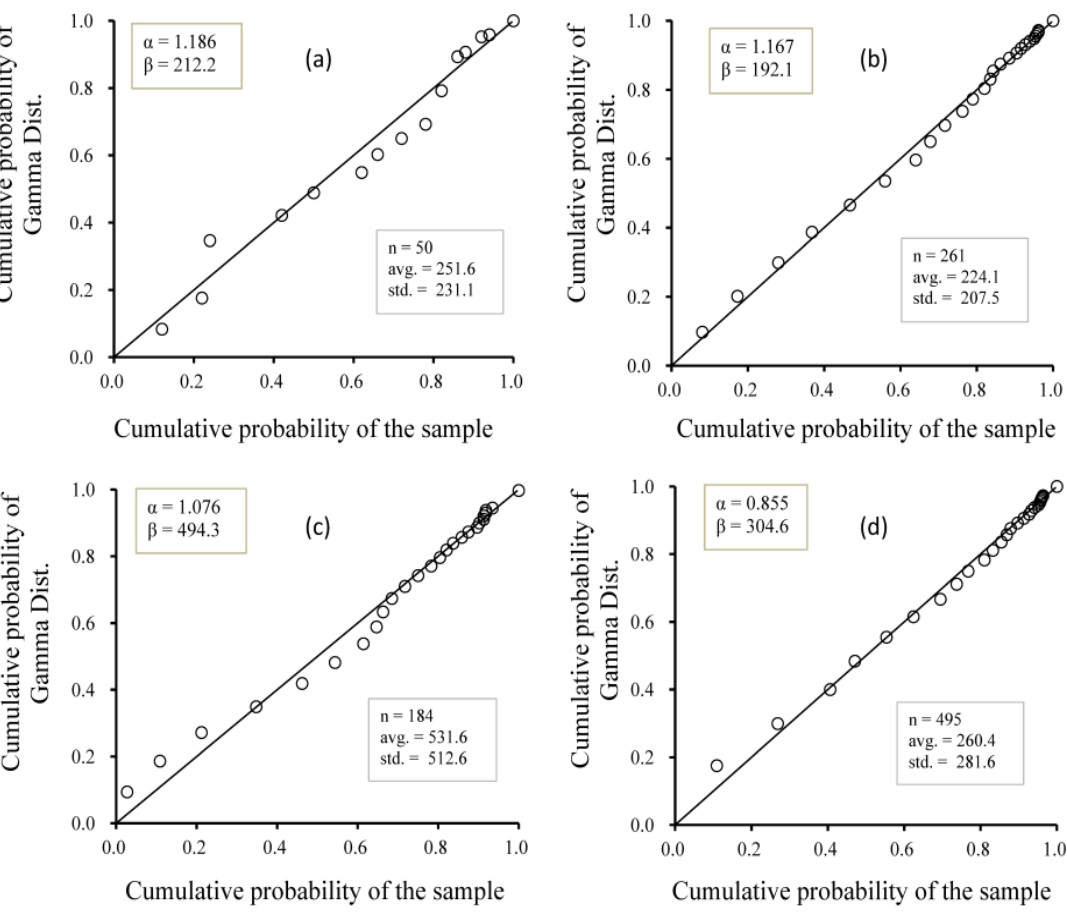





Figure 5

## Figure 6

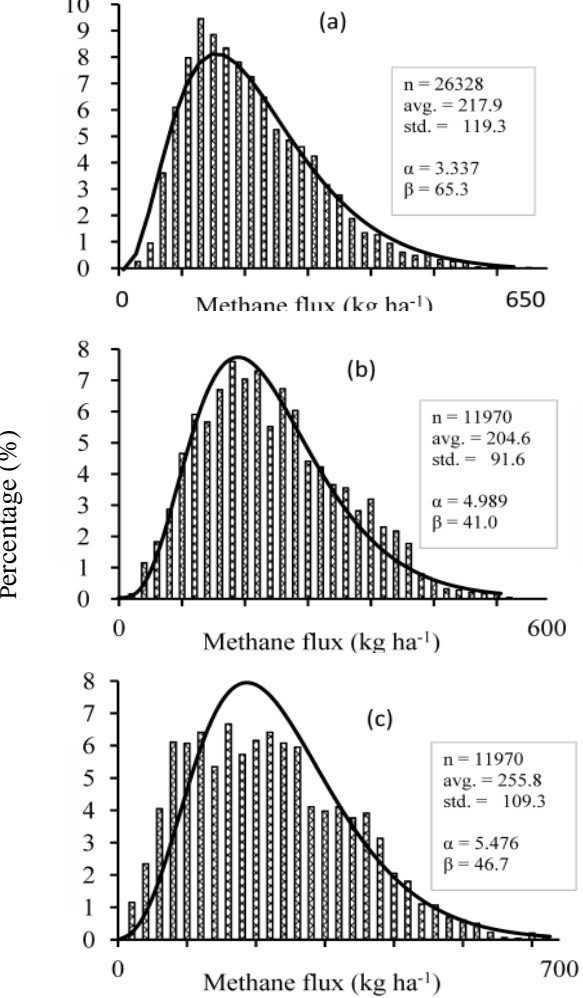



Figure 7

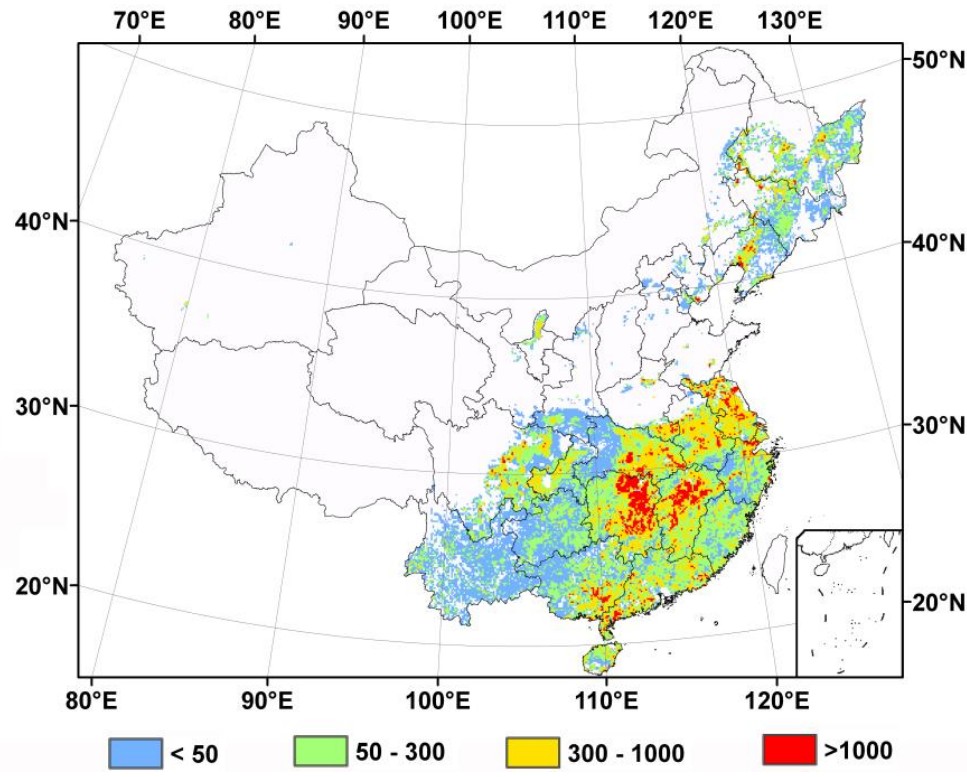



Figure 8

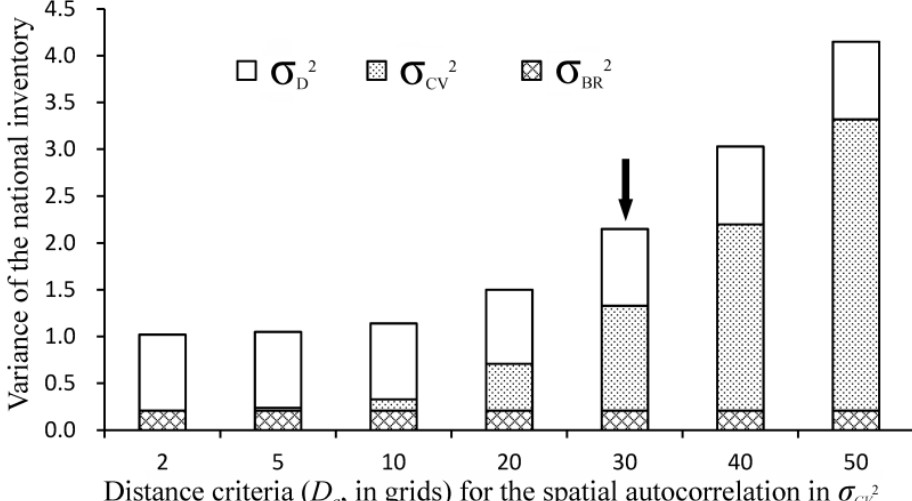