# Peer review of "Uncertainties in the national inventory of methane emissions from rice cultivation: field measurements and modeling approaches"

_Biogeosciences, 2016_

## Referee Comment (RC1) · Anonymous Referee #1 · 9 Sep 2016

The manuscript provides a comprehensive analysis of the sources of uncertainty in the national inventory of methane ($CH_4$) emissions from rice agriculture in China. Three approaches were used to estimate the inventory and the associated uncertainties (i.e. direct field measurements, two empirical regression models, and the process-based model, CH4MOD). Additionally, the sensitivity of the levels of uncertainty using each approach to various scenarios of data scarcity was assessed. The more complex, process-based model had the lowest total error compared to the two empirical models. All approaches had higher error when average values were used for input data compared to case-specific values, highlighting varying degrees of model instability to insufficiency of supporting data. Interestingly, even when no case-specific input data

were used in the processed-based model CH4MOD, it still had lower total error than the least complex empirical model when all case-specific input data were used (i.e. organic matter input modified by water regime). This in-depth comparison of approaches, their associated errors, and the sensitivity of the errors to input data availability is a significant contribution to the scientific community. It examines very relevant issues and challenges that modelers are faced with when scaling up field-validated models to larger spatial scales. The manuscript nicely quantifies and discusses the trade-offs associated with using the different approaches. It also outlines a method for assessing various sources of uncertainty and distinguishing model structural uncertainty from the uncertainty in input data.

The main criticisms I have of the paper are as follows:

-There is no mention of total estimated national CH4 emissions using each approach in the abstract. I actually I think the estimation of national CH4 emissions using the empirical models is missing from the whole paper. It seems like this is a major comparison to include in the paper and highlight in the abstract. Instead, the average CH4 emissions and 95% confidence intervals of the mean are reported. I think a comparison of the national CH4 emissions and their respective 95% confidence intervals for each approach and data-availability scenario is a very important application of this analysis and should be in the abstract. Similarly, I think it is important to highlight which case-specific data (e.g. organic matter inputs, water regime, or soil properties) mattered the most in terms of its effect on uncertainty when it was omitted.

-Overall the paper is lacking in citations of current research articles. Most articles cited are >10 years old.

-Missing description of model calibration of the two empirical models and CH4MOD. Thus, it's unclear whether data used for model validation (i.e. comparison to measurement-based estimations of fluxes) and uncertainty analyses are independent from data used to calibrate the internal model parameters.

-It's unclear whether the direct measurements used in the analyses are cumulative CH4 emissions or daily CH4 fluxes from the same experimental plots. If it's the latter, then the errors are not independent, and this issue should be explicitly addressed in the paper. The issue of non-independence of errors was discussed, but it was unclear whether this was due to measurements taken in close proximity versus repeatedly from the same location.

Additional comments, questions, and technical corrections:

P 1, Lines 26-29: Revise to account for the exception in which M-S3 performed better than R1-S0 (Table 2).

P 1, Line 33: Do you mean "between-grid variations", i.e. differences among grid cells?

P 2, Line 2: I think a slight rewording should be made, i.e. "Reducing the total uncertainty in the national methane inventory depends on a better understanding of both the complexity of the mechanisms of methane emission and the spatial correlations of the factors that influence methane emissions from rice paddies."

P3, Line 16: Reference needed.

P 6, Line 12: Provide detail on the parameters and assumptions for substrates produced from added organic matter and root exudates.

P 7, Line 9: Can you provide a reference or derivation of equation 4?

P 7, Line 10: Given that the focus of the manuscript is on uncertainty in national inventories, it seems that the methods section should be framed under national-level uncertainties as opposed to regional-level. It's my understanding that national inventories represent an aggregation of multiple regions. Thus, perhaps the section title here should state "national scales" as opposed to "regional scales", and translate this distinction into the text that follows.

P7, Lines 28-33: Please clarify each step of the process in which SAND data were obtained. What method of interpolation was used (e.g. ordinary kriging, inverse distance weighted)? What is meant by "missing spatial variation" in your dataset – how was this determined and quantified? Were some grid cells missing survey data all together?

P 9, Line 1: Please provide a reference or derivation of equation 5.

P 9, Line 9-10: You refer to the "three components of the estimation uncertainties" in equation 5. I assume you are referring to (1) (Fj x Br)2, (2) (Fj x CV)2, (3) $\sigma\_DJ^2$, which is analogous to the three terms in equation 6. Can you please provide a meaningful definition of what each of these components of uncertainty represent? Later in the discussion you explain that (Fj x Br)2 +(Fj x CV)2 represents model fallacy, while $\sigma\_DJ^2$ represents uncertainty due to input data. I think including this type of description in the methods section would be helpful to read leading into the results section.

P 9, Lines 30-31: Explicitly state the water regimes.

P 10, Line 27: What "estimated CH4 flux" are you referring to? Are you referring to an example of a single flux? If so, I would start the sentence with: "For example, in one case the modeled CH4 flux was . . ., while the measured flux was . . ."

P 11, Lines 16-18: Specify which model the simulated fluxes are based on. Please clarify this in Fig. 6 and Table 3 as well.

P 12, Lines 13-14: Didn't the authors also apply the two regression models to the 10 x 10 km grids? A comparison to the other two approaches (direct measurements and process-based model) should be discussed here.

P 12, Lines 26-29. Nice explanation!

P 14, Lines 19-33. See comment above for P 7, Line 10. Reframe conclusions to include national estimates and uncertainties at the broadest level of discussion.

---

## Referee Comment (RC2) · Anonymous Referee #2 · 15 Nov 2016

I agree with referee #1 that the paper 'Uncertainties in the national inventory of methane emissions from rice cultivation: field measurements and modeling approaches' by Zhang et al. is an important and nice study regarding general uncertainties evolving during regional/national GHG emission inventories. I also agree with referee #1 that national estimates of CH4 emissions should be more emphasized. My main criticism relates to the presentation of the study. Material and Methods, Results and Discussion sections all need revisions in order to improve the reader's access to the main points of this study (see specific comments).

Specific comments:

P1 L25: Mention that regression models are taken from literature.

P1 L27-28: Use clear measures and give respective values instead of using the vague term 'model performance' only.

P1 L30: Absolute values of simulated methane fluxes are meaningless here since context (e.g., different irrigation, straw management, ...) is not clear yet.

P4 L19-21: Statement is not very intuitive. Why should 'non-key' factors lead to significant errors? Factors leading to significant errors are implicitly named key.

P5 L21-22: Imprecise formulation, inaccuracies of models are manifold and should be defined more clearly based on common nomenclatures in literature, see for example nomenclature and definitions by (Kennedy and O'Hagan, 2001). Nomenclatures and definitions should be revised and standardized in many parts of the paper.

Kennedy, M.C., O'Hagan, A., 2001. Bayesian calibration of computer models. J. R. Stat. Soc. Ser. B Stat. Methodol. 63, 425–464. doi:10.1111/1467-9868.00294

P5: L34-36: Why were these two regression models chosen? It would be very interesting to see how IPCC emission factors, which also account for, e.g., different amounts of straw and different irrigation schemes would behave.

P6-7 Formulas 1-4: Unclear why these measures have been used. Give proper descriptions, meanings and references to 'bias' and 'total error' and compare both to each other.

P7 L12: ' errors in the performance of the method': unclear formulation, use consistent nomenclature for different error/uncertainty sources

P7 L 15: Give more information regarding your Monte Carlo simulation and PDFs since this is an important determinant of posterior uncertainty.

P8 L1 On what is this assumption (amount of stubble) based?

P8 L1-11: What is the difference between stubble and incorporated straw?

P8 L15-16: Be more precise here and mention considered irrigation schemes and how the model handles them.

P8 L28: Probably Appendix B is meant.

P8 L33: In section 2.4, the description of used formulas should be improved since the combination of model and model input uncertainty is a central point of this study. The derivations of formulas in the Appendix are unclear. Give consistent names and meanings to each symbol that is used. Parts in the Discussion sections refer to the meaning of formulas and measures and should be moved here.

P9 L27: Do you mean 'harvested-area-weighted' or 'cultivated-area-weighted'? Since cropping intensity (number of crops per year) varies, the weighted mean should be derived based on harvested area. In addition to area weighted means you should also consider seasonal means. A given amount of data may refer to different seasons, e.g., winter, spring, summer and autumn with strongly varying potentials of CH4 emissions. Most likely the seasonality distribution of observations does not correspond to the actual seasonality distribution of rice cultivation in China.

P10 L18-21: Be more precise how measurements are dependent from each other. The potential dependency of measurements is not discussed in the Discussion section.

P10 L12-16: Standard Error (SE) and deviation are very common measures and do not need explanations/references. To my understanding, the presented SE refers to the variability of different observed mean fluxes from different field sites. How are measurement errors reflected? What do you mean with representative error?

P10 L25: Present average values of overestimations for both models.

P10 L35: Why is 'total error' and not 'bias' interpreted as model performance? In order to underline this statement, more measures should be used, e.g., root mean squared error, R2, model efficiency.

P11 L22: I miss the discussion of these values. Are such uncertainties small or large

compared to other studies?

P11 L23-25: Discussion is missing.

P11 L30 - P12 L14: This is rather introduction and representing of results than discussion.

P12 L10: Temporal variations are not presented.

P12 L15-17: Unclear argumentation.

P12 L18-22: Unclear argumentation. Model performance was assessed with site-specific input and not with regional averages. The representation of experimental measurements for larger regions and associated uncertainties should be independent of models. Discussion of comparison between model and measurements at site scale could be moved to a separate subsection.

P12 L31-33: Should be moved to the Results section. Use consistent nomenclature, i.e., the term 'model fallacy' has not been used beforehand. Do not repeat formulas from the Material and Methods section in the Discussion.

P12 L35 - P13 L14: Much of this information belongs to the Material and Method Section and to the Discussion. Key results (e.g., '56.6% of total uncertainty originates from the model'), which are also presented in the abstract should be first presented in the Results section and subsequently discussed. Appropriate discussion regarding the different uncertainty sources (model versus input) is missing. Argumentation regarding 'imprecision random noise and/or unknown factors' is unclear.

P13 L27 - P14 L16: Remove this section from the Discussion. This is partly Material and Methods and seems to be an arbitrary example of model parameter uncertainty that has been neglected and thus is not much contributing to this study.

Fig. 5: Use identical axes for all plots.

[Figure]

---

## Author Comment (AC1) · 6 Dec 2016

Reviewer #1: Comments

The manuscript provides a comprehensive analysis of the sources of uncertainty in the national inventory of methane ($CH_4$) emissions from rice agriculture in China. Three approaches were used to estimate the inventory and the associated uncertainties (i.e. direct field measurements, two empirical regression models, and the process-based model, CH4MOD). Additionally, the sensitivity of the levels of uncertainty using each approach to various scenarios of data scarcity was assessed. The more complex, process-based model had the lowest total error compared to the two empirical models. All approaches had higher error when average values were used for input data

compared to case-specific values, highlighting varying degrees of model instability to insufficiency of supporting data. Interestingly, even when no case-specific input data were used in the processed-based model CH4MOD, it still had lower total error than the least complex empirical model when all case-specific input data were used (i.e. organic matter input modified by water regime). This in-depth comparison of approaches, their associated errors, and the sensitivity of the errors to input data availability is a significant contribution to the scientific community. It examines very relevant issues and challenges that modelers are faced with when scaling up field-validated models to larger spatial scales. The manuscript nicely quantifies and discusses the trade-offs associated with using the different approaches. It also outlines a method for assessing various sources of uncertainty and distinguishing model structural uncertainty from the uncertainty in input data. Re: We greatly appreciate the reviewer's comments on the scientific significance of the study.

-There is no mention of total estimated national CH4 emissions using each approach in the abstract. I actually I think the estimation of national CH4 emissions using the empirical models is missing from the whole paper. It seems like this is a major comparison to include in the paper and highlight in the abstract. Instead, the average CH4 emissions and 95% confidence intervals of the mean are reported. I think a comparison of the national CH4 emissions and their respective 95% confidence intervals for each approach and data-availability scenario is a very important application of this analysis and should be in the abstract. Similarly, I think it is important to highlight which case specific data (e.g. organic matter inputs, water regime, or soil properties) mattered the most in terms of its effect on uncertainty when it was omitted. Re: Many thanks to the suggestion that "a comparison of the national CH4 emissions and their respective 95% confidence intervals for each approach and data-availability scenario is a very important application of this analysis and should be in the abstract". In the revision, we have made the comparison of the national CH4 emissions and their respective 95% confidence intervals for each approach and data availability scenarios. The results of the comparison were showed in Table 3 and the description of the results was also added in the main text

(P13 lines 3-12 in the 'clean revised manuscript'). In Table 3, the estimated national CH4 emissions ranged from 6.43 (3.79‒9.77) Tg to 13.59 (1.45‒38.98) Tg for the M-S0 scenario R1-S0 scenario, respectively. The 95% CIs of the national estimation differed more greatly among the approaches than those among the data availability scenarios of each approach. As an indicator of the trade-off between the complexity of the approach and data availability, the $\sigma d/\sigma b+v$ ratio in Table 3 was 0.87 for M-S0, closer to 1 than those for the other approaches and scenarios, which also yielded the narrowest 95% CI in Table 3. The factors affecting methane emission from rice paddies (e.g. organic matter inputs, water regime, or soil properties) had been incorporated into CH4MOD as input variables. The importance of those factors on uncertainty had been discussed in a previous study (Zhang et al., 2014). Stating briefly, the factor of high sensitivity will result in larger uncertainty when omitted, from water regime down to soil properties and organic matter inputs. As suggested by the reviewer, we also add statement of the total estimated national CH4 emissions in the revised abstract (P1 lines 28-29).

-Overall the paper is lacking in citations of current research articles. Most articles cited are >10 years old. Re: The topic of the present study, uncertainties in the modelling approaches closely related to methane emissions from rice paddies and the relevant, had been dedicatedly discussed in few previous studies (Ogle, et al., 2010; van Bodegom et al, 2002a). In the present study, we compared performances of CH4MOD and two empirical methods that had been developed and utilized in early days (Neue et al., 1990; Khalil et al., 1991, 1993; Bachelet et al., 1995; Kern et al., 1995, 1997), and had to reach out to studies 10-20 years ago. We, however, didn't omit relevant studies in recent years, e.g., the study of mitigating methane emission from rice cultivation by gene transcription (Su et al., Nature, 2015), the study of methanogenic community structure involving methane production (Singh et al., SBB, 2012), and national/global estimation of methane emissions from rice paddies and wetlands (Chen et al., GCB, 2013; Ren et al., Tellus B, 2011; Zhang et al., GCB, 2011). In the revision, we referenced major results of the recent studies concerning methane emission from rice paddies (Ito et al.,

2012; Tian et al., 2016; Weller et al., 2016; Zhang et al., 2016; Dijkstra et al., 2012).

Missing description of model calibration of the two empirical models and CH4MOD.Thus, it's unclear whether data used for model validation (i.e. comparison to measurement-based estimations of fluxes) and uncertainty analyses are independent from data used to calibrate the internal model parameters. Re: The approaches in the study had been used in previous studies (Bachelet et al., 1995; Kern et al., 1995, 1997; Zhang et al., 2011) to estimated methane emissions from rice paddies on regional, national and global scales. When analyzing the performances of the approaches in the present study, we validated them with data excluding those had been used for calibration to maintain the independence between the validation and calibration. We explicitly addressed the situation in the revised MS (P5 Lines 23-24).

It's unclear whether the direct measurements used in the analyses are cumulative CH4 emissions or daily CH4 fluxes from the same experimental plots. If it's the latter, then the errors are not independent, and this issue should be explicitly addressed in the paper. The issue of non-independence of errors was discussed, but it was unclear whether this was due to measurements taken in close proximity versus repeatedly from the same location. Re: All the measurements of CH4 emission in the present study are cumulative CH4 emissions over the period from rice transplanting to harvesting. We explicitly stated it in the revised MS (P10 Lines 13-14). We discussed the non-independence of the measurements due to spatially close proximity in Section 4.1, when no temporal dependence of the daily measurements involved.

Additional comments, questions, and technical corrections: P 1, Lines 26-29: Revise to account for the exception in which M-S3 performed better than R1-S0 (Table 2). Re: We revised the sentence as "Comparisons revealed that the CH4MOD model may perform worse than the comparatively simple regression models when no sufficient input data for the model were available".

P 1, Line 33: Do you mean "between-grid variations", i.e. differences among grid

cells? Re: It is the within-grid variation calculated via the Monte-Carlo method. To make it clearer, we revised the sentence as "the within-grid variations, $\sigma T_{,,}i$, were found to be 81.2%–95.5% to the grid cell means (Fi)."

P 2, Line 2: I think a slight rewording should be made, i.e. "Reducing the total uncertainty in the national methane inventory depends on a better understanding of both the complexity of the mechanisms of methane emission and the spatial correlations of the factors that influence methane emissions from rice paddies." Re: Thanks for the suggestion. The sentence was revised.

P3, Line 16: Reference needed. Re: The appropriate literature references were added.

P 6, Line 12: Provide detail on the parameters and assumptions for substrates produced from added organic matter and root exudates. Re: We added sentences to briefly describe the substrate production from added organic matter and root exudates in the revised Supporting Information (Appendix B). The amount of the substrate derived from rice root exudate was simulated by a power function of the rice biomass, scaled by the parametric influence of the soil context and the rice cultivar. The substrate derived from the added organic matter was calculated by a first-order kinetic decomposition equation of the organic matter in soil, also scaled by the parametric influence of the soil context and the temperature. Details can be found in Huang et al (2004).

P 7, Line 9: Can you provide a reference or derivation of equation 4? Re: We detailed the derivation of Equation 4 in the revision.

P 7, Line 10: Given that the focus of the manuscript is on uncertainty in national inventories, it seems that the methods section should be framed under national-level uncertainties as opposed to regional-level. It's my understanding that national inventories represent an aggregation of multiple regions. Thus, perhaps the section title here should state "national scales" as opposed to "regional scales", and translate this distinction into the text that follows. Re: Thanks for the suggestion. We revised it

throughout the section and other places in the MS.

P7, Lines 28-33: Please clarify each step of the process in which SAND data were obtained. What method of interpolation was used (e.g. ordinary kriging, inverse distance weighted)? What is meant by "missing spatial variation" in your dataset – how was this determined and quantified? Were some grid cells missing survey data all together? Re: Soil properties have extremely high spatial variation and may vary largely from one place not far from another. We obtained the data from Institute of Soil Sciences, Chinese Academy of Sciences, as indicated in the MS. They collected more than 7000 soil profile measurements sampled during the period from 1980s to the present and linked them to the a soil database of 1:1,000,000 scale (Shi et al., 2004), and produced the gridded data of soil properties with geostatistical methods. We compared the spatial variation explained in the gridded datasets of soil properties against the variations in the profile measurements to analyze the 'missing spatial variation' (Bodegom et al., 2002b). The 'missing spatial variation' is the proportion of spatial variation of the soil properties (the sand content of the surface soil layer in the present study) that were not accounted for by the gridded datasets. We used the missing variation to build the PDF of SAND in Monte Carlo simulation by assuming normal distributions of the missing variation. We added the brief description of the soil property datasets in Appendix B.

P 9, Line 1: Please provide a reference or derivation of equation 5. Re: Equation 5 is derived from Equation (C9) in the Supporting Information (Appendix C) and Equation 4 in the main text, when used in each grid cell. We added the description and derivation in revision (P9 Lines 27-30 and P10 Lines 1-10).

P 9, Line 9-10: You refer to the "three components of the estimation uncertainties" in equation 5. I assume you are referring to (1) $(F_j \times B_r)^2$, (2) $(F_j \times CV)^2$, (3) _DJËĘ2, which is analogous to the three terms in equation 6. Can you please provide a meaningful definition of what each of these components of uncertainty represent? Later in the discussion you explain that $(F_j \times B_r)^2 + (F_j \times CV)^2$ represents model fallacy, while

_DJËȨ2 represents uncertainty due to input data. I think including this type of description in the methods section would be helpful to read leading into the results section. Re: Thank for your suggestion. We added explanation of the terms in the revised MS in the method section. In Equation 5, signifies the uncertainty caused by the error and availability of data, $(Fi{\times}rb)2$ represents the modelling bias, and $(Fi{\times}rv)2$ represents the rest parts of the model fallacy error apart from $(Fi{\times}rb)2$. We provided more details of the derivation and explanation of Equation 5 in the Supporting Information (Appendix C) in more general terms than the main text. The three components in Equation 5 do correspond to those in Equation 6 and the derivation from Equation 5 to Equation 6 was also provided in the Supporting Information (Section D).

P 9, Lines 30-31: Explicitly state the water regimes. Re: Revised.

P 10, Line 27: What "estimated CH4 flux" are you referring to? Are you referring to an example of a single flux? If so, I would start the sentence with: "For example, in one case the modeled CH4 flux was . . ., while the measured flux was . . ." Re: Thank you. We revised the sentence (P11 Lines 28-30).

P 11, Lines 16-18: Specify which model the simulated fluxes are based on. Please clarify this in Fig. 6 and Table 3 as well. Re: Thank you for the comments. We added information to specify the model used (P12 Lines 19-20) and the caption of Fig. 6.

P 12, Lines 13-14: Didn't the authors also apply the two regression models to the 10 x 10 km grids? A comparison to the other two approaches (direct measurements and process-based model) should be discussed here. Re: The two regression models were not used to the 10 x 10 km grid in the BGD version of the MS. As suggested by the reviewer, we applied all the three models and data availability scenarios in the revision and list the results in the revised Table 3, focusing on the national CH4 emission and the relevant uncertainties. While there are no measurements on grids, comparison of the estimation by modelling can only be carried among the models (Table 3 and P13, Lines 3-12) instead of against measurements.

P 12, Lines 26-29. Nice explanation! Re: Thank you.

P 14, Lines 19-33. See comment above for P 7, Line 10. Reframe conclusions to include national estimates and uncertainties at the broadest level of discussion. Re: We revised the conclusion and discussion section and added information of the national estimates in both the abstract and the conclusion.

Please also note the supplement to this comment:
http://www.biogeosciences-discuss.net/bg-2016-250/bg-2016-250-AC1-supplement.pdf

---

## Author Comment (AC2) · 6 Dec 2016

I agree with referee #1 that the paper 'Uncertainties in the national inventory of methane emissions from rice cultivation: field measurements and modeling approaches' by Zhang et al. is an important and nice study regarding general uncertainties evolving during regional/national GHG emission inventories. I also agree with referee #1 that national estimates of CH4 emissions should be more emphasized. My main criticism relates to the presentation of the study. Material and Methods, Results and Discussion sections all need revisions in order to improve the reader's access to the main points of this study (see specific comments).

Re: We thank the reviewer for the comments and made revision to the MS to show the

national estimations of CH4 emission, and also the writing of the MS emphasizing the formulation and nomenclatures.

Specific comments: P1 L25: Mention that regression models are taken from literature.

Re: Revised (P1 Line18 in the 'clean revised manuscript').

P1 L27-28: Use clear measures and give respective values instead of using the vague term 'model performance' only.

Re: The 'model performance' refers to how the model representing the variation in the observations, evaluated by the difference between the observations and the corresponding model outputs. Conceptually, the model performance here covers the 'parameter uncertainty' and 'model inadequacy' in Kennedy and O'Hagan (2001) and errors in observations, because we can't distinguish them with model validation, which was used to evaluate the 'model performance' in the present study. The measures to quantify the model performance here are two statistical parameters of the modelling residuals (difference between the observations and modelling outputs): bias (means of the residuals) and variance (statistical variance of the residuals) as showed in Equation 1 and Equation 2. We revised the equation and the main text to make the meaning of the terms clearer.

P1 L30: Absolute values of simulated methane fluxes are meaningless here since context (e.g., different irrigation, straw management, ...) is not clear yet.

Re: The modelling result here is the result of CH4MOD with available information of irrigation, straw management and soil properties of paddies in rice cultivation of China. We revised the sentence as 'As simulated by CH4MOD with data of irrigation, organic matter incorporation and soil properties of rice paddies, the modelling methane fluxes varied from 17.2 kg CH4 ha–1 to 708.3 kg CH4 ha–1'

P4 L19-21: Statement is not very intuitive. Why should 'non-key' factors lead to significant errors? Factors leading to significant errors are implicitly named key.

[Figure]

Re: Here we intended to say that they were not 'non-key' at all. To avoid misunderstanding, we revised the sentence by replacing 'non-key' with 'other'. P5 L21-22: Imprecise formulation, inaccuracies of models are manifold and should be defined more clearly based on common nomenclatures in literature, see for example nomenclature and definitions by (Kennedy and O'Hagan, 2001). Nomenclatures and definitions should be revised and standardized in many parts of the paper. Kennedy, M.C., O'Hagan, A., 2001. Bayesian calibration of computer models. J. R. Stat. Soc. Ser. B Stat. Methodol. 63, 425–464. doi:10.1111/1467-9868.00294 Re: We thank the reviewer for the comments and recommending the literature. Here, the model inaccuracy refers to the combination of 'model inadequacy' and 'parameter uncertainty' in Kennedy and O'Hagan (2001). In other places of the MS, 'model fallacy' means the same. We revised the MS to use the term 'model fallacy' throughout the MS and explain explicitly the mean of it. We also rewrite the equations of the MS.

P5: L34-36: Why were these two regression models chosen? It would be very interesting to see how IPCC emission factors, which also account for, e.g., different amounts of straw and different irrigation schemes would behave.

Re: One of the objectives of the study was to compare the performance of models with different complexity with different levels of data availability. We chose the two regression models because: 1) they had been used to estimate regional/national/global methane missions in many previous studies, and 2) they differed from each other and from CH4MOD explicitly in levels of complexity. There are many other models that developed and used widely in modeling methane emissions from rice paddies and wetlands etc. But we can't tell which one is more complicated in structure that the other. We briefly explained it in the revise MS (P6 Lines 13-15).

P6-7 Formulas 1-4: Unclear why these measures have been used. Give proper descriptions, meanings and references to 'bias' and 'total error' and compare both to each other.

Re: 'bias' is the statistical mean of the modelling residuals. We admit that 'total error' is not a proper term. In the MS, it means the 'mean of squared errors' in model validation. In the revised MS, we used the term 'mean of squared errors' in the main text.

P7 L12: ' errors in the performance of the method': unclear formulation, use consistent nomenclature for different error/uncertainty sources

Re: revised as 'model fallacy'. Because in the MS, our emphasis was on the quantification of the uncertainty in the national inventory by modelling approaches, the rationale of the uncertainty was mainly provided in the Supporting Information (Appendix C and D).

P7 L 15: Give more information regarding your Monte Carlo simulation and PDFs since this is an important determinant of posterior uncertainty.

Re: To measure the uncertainties in model outputs due to insufficient data quality and availability, we applied Monte Carlo simulations to the CH4MOD model. Statistical characteristics were derived from the available datasets to develop probability distribution functions (PDFs) for each model input variable. The PDF of field irrigation were defined by the occurrence percentage of each irrigation pattern (Table B2). Table B1 shows the statistical parameters of the PDF (normal distribution) of organic matter incorporation in each province. The PDF of the soil sand percentage was also built as normal distribution with parametric information from the literature (Shi et al., 2004). We performed Monte Carlo simulation in the way of randomly drawing values of the model input variables from their PDFs and then run the model (e.g. CH4MOD). This process iterated 1000 times and at the last step, the mean and 95% CI of the calculated methane fluxes were derived from the iterations (P8 Lines 3-6).

P8 L1 On what is this assumption (amount of stubble) based? P8 L1-11: What is the difference between stubble and incorporated straw?

Re: Stubble is the part of rice stem that left after rice harvesting. Traditionally, both the

rice grain and rice straw were harvested and stubble was left in field. The harvested straw may be taken away or left in field, but stubble was always left. The amount of stubble accounts for about 10% of the aboveground biomass of rice according to previously published literatures (Huang et al., 2004; Zhang et al., 2011). We noted the literature in the revised MS (P8 Line24).

P8 L15-16: Be more precise here and mention considered irrigation schemes and how the model handles them.

Re: The irrigation in rice cultivation in China was summarized into five patterns: 1) flooding-drainage-flooding-intermittent irrigation, 2) flooding-drainage-intermittent irrigation, 3) flooding-intermittent irrigation, 4) continuous flooding and 5) continuously intermittent irrigation (Gao and Li, 1992; Huang et al., 2004). Appendix B in the Supporting Information provides necessary information of the irrigation in China. Table B2 list the percentage of each water pattern in different regions of China. More information of how CH4MOD handles the irrigation may refer to the literature of the model development (Huang et al., 2004). We also add brief description of the irrigation in the revised MS (P8 Lines 32-34 and P9 Line 1).

P8 L28: Probably Appendix B is meant.

Re: Revised.

P8 L33: In section 2.4, the description of used formulas should be improved since the combination of model and model input uncertainty is a central point of this study. The derivations of formulas in the Appendix are unclear. Give consistent names and meanings to each symbol that is used. Parts in the Discussion sections refer to the meaning of formulas and measures and should be moved here.

Re: We revised the relevant part of the MS and the Appendix, emphasizing the consistence of the names and expressions.

P9 L27: Do you mean 'harvested-area-weighted' or 'cultivated-area-weighted'? Since

cropping intensity (number of crops per year) varies, the weighted mean should be derived based on harvested area. In addition to area weighted means you should also consider seasonal means. A given amount of data may refer to different seasons, e.g., winter, spring, summer and autumn with strongly varying potentials of CH4 emissions. Most likely the seasonality distribution of observations does not correspond to the actual seasonality distribution of rice cultivation in China.

Re: Yes. The 'area-weighted' in the MS means 'harvested-area-weighted' and we revised the expression.

In China, the rice cultivation is different from north to south: single rice cultivation in north-eastern China, rice-upland crop rotation in eastern China and double rice cultivation in southern China. The 'harvested-area-weighted' analysis in the present study distinguished the harvested area of different water irrigation, because irrigation is the most important factor for methane emission. Seasonality also affects the methane emission but not as important as irrigation, according to both observational and modelling studies (Yan et al., 2005; Zhang et al., 2011). We agree with the reviewer that mismatch between the seasonality of the observations and the actual rice cultivation in China may bias the national estimation of CH4 emission via the statistical summation of the observations, and contributes to the uncertainty of the estimation.

P10 L18-21: Be more precise how measurements are dependent from each other. The potential dependency of measurements is not discussed in the Discussion section.

Re: The dependence of measurements here means the possible spatial correlation among them because of the common environmental conditions they may share. It is not the meaning that they were dependently obtained by sampling. The spatial aggregation of the measurements obtained at different places to produce national estimations may introduce biases if the spatial correlation among the measurements were not handled properly. We didn't make in-depth discussion about the spatial correlation because it is beyond the topic of the present study. In the revision, we revised the sentence to clear

that it is about the spatial correlation. We also provide literature reference (Legendre, 1993; Dormann et al., 2007) for those interested in spatial correlation.

P10 L12-16: Standard Error (SE) and deviation are very common measures and do not need explanations/references. To my understanding, the presented SE refers to the variability of different observed mean fluxes from different field sites. How are measurement errors reflected? What do you mean with representative error?

Re: Yes, we agree with you concerning the standard error. The measurement errors were not discussed separately in the present study. The reason of doing so was explained in Appendix C. The representative error in the present and other literatures (e.g., Van Bodegom et al., 2002a; Verburg et al., 2006) stands for the representativeness of the measurement obtained at a site to the area that enclose the site.

P10 L25: Present average values of overestimations for both models.

Re: Revised.

P10 L35: Why is 'total error' and not 'bias' interpreted as model performance? In order to underline this statement, more measures should be used, e.g., root mean squared error, R2, model efficiency.

Re: 'bias' is the average of the modelling residuals, accounting part of the errors. We use 'mean of the squared errors' to interpret model performance. 'total error' is not a proper expression and we replace it with 'mean of the squared errors' in the revision. There are other indexes, e.g., R2 and RMSE, we use bias and 'mean of the square errors' in the MS owing to they are directly comparable to the errors from data availability.

P11 L22: I miss the discussion of these values. Are such uncertainties small or large compared to other studies?

Re: The within-grid estimation error ($\sigma$T,i, calculated with Equation 5) is the error in each grid cell due to both the model fallacy and data scarcity when making estimation

of a grid cell (10 × 10km). They are not shown in details because we emphasized the uncertainty in the national inventory, which was the spatial aggregation of the uncertainty in each grid cells. We didn't compare the result of the 'within-grid estimation error' in the present study with other studies because no study had make estimation of the uncertainty in the way of the present study.

P11 L23-25: Discussion is missing.

Re: In the revision, we discussed the difference of the national methane emissions and the uncertainties estimated with different approaches and the data availability scenarios, as showed in the revised Table 3.

P11 L30 - P12 L14: This is rather introduction and representing of results than discussion.

Re: Thanks for this comment. We revised the MS by moving it to the introduction.

P12 L10: Temporal variations are not presented.

Re: Here in the sentence, we noted that there are temporal variations, annual, seasonal and even diurnal, in the methane emissions. But in the present study, we discussed the spatial variation and the estimation uncertainty in the national inventory of a specific year. Temporal variations of the methane emission were not discussed.

P12 L15-17: Unclear argumentation.

Re: Thank you for pointing it out. We revised the sentence as 'This was partly due to the discrepancy in the spatial representativeness of the methane fluxes in field observations and model estimations'.

P12 L18-22: Unclear argumentation. Model performance was assessed with site-specific input and not with regional averages. The representation of experimental measurements for larger regions and associated uncertainties should be independent of models. Discussion of comparison between model and measurements at site scale

could be moved to a separate subsection.

Re: This is what the 'representative error' means, which had been discussed in previous studies (Verburg et al., 2006; Van Bodegom et al., 2002a) and described in Appendix C of the MS. Model performance was assess with site-specific input. Here the 'site' means a small scale (e.g., a hectare or smaller) instead of a 'point', when the experimental sampling was taken at several 'points' called 'duplicates' at the experimental site. When we use the model for regional estimation, we make estimations for each grid cell (10 $\times$ 10 km in the present study). The mismatch of the scale supports the meaning of 'representative error'.

P12 L31-33: Should be moved to the Results section. Use consistent nomenclature, i.e., the term 'model fallacy' has not been used beforehand. Do not repeat formulas from the Material and Methods section in the Discussion.

Re: Thanks for the suggestion, we revised accordingly (P12 Lines 24-26).

P12 L35 - P13 L14: Much of this information belongs to the Material and Method Section and to the Discussion. Key results (e.g., '56.6% of total uncertainty originates from the model'), which are also presented in the abstract should be first presented in the Results section and subsequently discussed. Appropriate discussion regarding the different uncertainty sources (model versus input) is missing. Argumentation regarding 'imprecision random noise and/or unknown factors' is unclear.

Re: Section 4.1 discussed the different error sources to the uncertainties in the inventory. This paragraph around Fig. 8 was about the aggregation of $\sigma$v,i2. Material and Method Section described how the errors were quantified and aggregated, as showed in Fig. 2. We thank the reviewer for the revision suggestion and revised the MS accordingly.

P13 L27 - P14 L16: Remove this section from the Discussion. This is partly Material and Methods and seems to be an arbitrary example of model parameter uncertainty

that has been neglected and thus is not much contributing to this study.

Re: Section 4.2 discuss how model improvement (e.g., parameterizing rice cultivar more specifically) affect the uncertainty analysis. We agree with the reviewer that the model parameter uncertainty wan not separately analyzed in present study. But because the parameter uncertainty contributed significantly to the model fallacy, it should be noted briefly in the discussion.

Fig. 5: Use identical axes for all plots.

Re: We guess you meant Fig. 5. We had at first used identical axes for Fig. 5-(a), Fig. 5-(b) and Fig. 5-(c). But it looked a little awkward, we, therefore, changed the y-axe of Fig. 5-(a) and kept the other axes identical.

Please also note the supplement to this comment:
http://www.biogeosciences-discuss.net/bg-2016-250/bg-2016-250-AC2-supplement.pdf